# Distant relatives of a eukaryotic cell-specific toxin family evolved a complement-like mechanism to kill bacteria

Hunter L. Abrahamsen[1,8], Tristan C. Sanford[1,8], Casie E. Collamore[1,8], Bronte A. Johnstone [2,3,8], Michael J. Coyne[4], Leonor García-Bayona[4], Michelle P. Christie [2,3], Jordan C. Evans[1,6], Allison J. Farrand[1,6], Katia Flores[4], Craig J. Morton [2,7], Michael W. Parker [2,3,5] ✉, Laurie E. Comstock [4] ✉ & Rodney K. Tweten [1] ✉

Cholesterol-dependent cytolysins (CDCs) comprise a large family of pore-forming toxins produced by Gram-positive bacteria, which are used to attack eukaryotic cells. Here, we functionally characterize a family of 2-component CDC-like (CDCL) toxins produced by the Gram-negative Bacteroidota that form pores by a mechanism only described for the mammalian complement membrane attack complex (MAC). We further show that the *Bacteroides* CDCLs are not eukaryotic cell toxins like the CDCs, but instead bind to and are proteolytically activated on the surface of closely related species, resulting in pore formation and cell death. The CDCL-producing *Bacteroides* is protected from the effects of its own CDCL by the presence of a surface lipoprotein that blocks CDCL pore formation. These studies suggest a prevalent mode of bacterial antagonism by a family of two-component CDCLs that function like mammalian MAC and that are wide-spread in the gut microbiota of diverse human populations.

The mechanism of pore formation by the cholesterol-dependent cytolysins (CDC)[1–3] established a paradigm for the understanding of the assembly of a wide variety of prokaryotic[4] and eukaryotic β-barrel pore-forming toxins and immune defense proteins[5]. The CDCs are well-known virulence factors that are mostly restricted to Gram-positive bacterial pathogens, which bind to and form large β-barrel pores in cholesterol-rich eukaryotic membranes[6]. We previously reported that *Elizabethkingia anophelis*, a bacterial species of the phylum Bacteroidota and a commensal of the malarial mosquito midgut produces a 2-component CDC-like

(CDCL) set of proteins, encoded by adjacent genes: one large (CDCL$^L$) and one smaller component (CDCL$^S$)[7]. The CDCL$^L$ subunit lacks the characteristic signature features for binding cholesterol, and the primary structure of the CDCL$^S$ subunit completely lacks an identifiable receptor-binding domain. The mechanism of their action and their in vivo targets are unknown, as purified proteins analyzed on lipid bilayers exhibited weak and infrequent in vitro pore-forming activity.

The studies herein show that, unlike their CDC relatives, the CDCLs require proteolytic activation of two distinct components that

[1]Department of Microbiology & Immunology, The University of Oklahoma Health Sciences Center, Oklahoma City, OK, USA. [2]Department of Biochemistry and Pharmacology, Bio21 Molecular Science and Biotechnology Institute, University of Melbourne, Parkville, VIC 3010, Australia. [3]ARC Centre for Cryo-electron Microscopy of Membrane Proteins, Bio21 Molecular Science and Biotechnology Institute, University of Melbourne, Parkville, VIC 3010, Australia. [4]Duchossois Family Institute and Department of Microbiology, University of Chicago, Chicago, IL, USA. [5]Australian Cancer Research Foundation Rational Drug Discovery Centre, St Vincent's Institute of Medical Research, Fitzroy, VIC 2065, Australia. [6]Present address: Wheeler Bio, Oklahoma City, OK 73104, USA. [7]Present address: CSIRO Biomedical Manufacturing Program, Clayton, VIC 3168, Australia. [8]These authors contributed equally: Hunter L. Abrahamsen, Tristan C. Sanford, Casie E. Collamore, Bronte A. Johnstone. ✉e-mail: mwp@unimelb.edu.au; lecomstock@uchicago.edu; rod-tweten@ouhsc.edu

then interact and form a pore to kill bacteria using a mechanism only described for the mammalian complement membrane attack complex (MAC). Genomic and metagenomic analysis reveals that the CDCLs are widespread in major species of Bacteroidales that inhabit the human gut, with some widely distributed among these species on mobile genetic elements (MGEs). These studies establish a new paradigm by which a bacterial pore is assembled, show that a distant phylogenetic branch of the CDCs kills bacteria rather than eukaryotic cells, and reveal genes for the CDCLs are distributed in the gut microbiota of geographically widespread human populations.

## Results

### Identification and analysis of CDCL pairs in Bacteroidota genomes

Our previous preliminary analysis of the distribution of CDC-like proteins in phylogenetically diverse bacterial species revealed that many Bacteroidota genomes encode CDCL proteins[7]. Here, we comprehensively analyzed the genomes of 5947 Bacteroidota strains, querying for proteins with PF01289.22, a motif in diverse CDC and MACPF proteins, and identified 2447 full-length proteins from 1578 genomes, with most genomes encoding a single gene (821 genomes), two adjacent genes (675 genomes) or three adjacent CDCL encoding genes (61 genomes). To begin to understand the role of dual CDCL proteins identified in *E. anophelis*, we limited the set to pairs or triplets of proteins encoded by adjacent genes with predicted SpII signal peptides allowing surface localization and secretion[8]. These criteria reduced the list to 1120 CDCL proteins from 497 genomes including 8 families, 12 genera, and 36 Bacteroidota species (Supplementary Dataset 1). Grouping of proteins at 96% identity resulted in binning into 54 CDCL clusters (Supplementary Dataset 1, tab 1) with 28 distinct CDCL patterns (pairs or triplets, Supplementary Dataset 1, tab 2). CDCL pairs were of two distinct types: (1) the protein encoded by the upstream gene is smaller than the second gene, ~365 aa for the small subunit (CDCL$^S$) and ~500 aa for the large subunit (CDCL$^L$), or (2) the proteins encoded by adjacent genes are of similar size but range from 336 aa to 621 aa. A phylogenetic tree of the 54 CDCL proteins revealed that the CDCL$^S$ and CDCL$^L$ subunits segregate to separate branches; with a third branch comprised of the 6 pairs of similar-sized CDCLs (Fig. 1a). Each similar-sized CDCL is most related to its paired protein. Each of the 3-component CDCL proteins segregated to three distinct branches.

Each pattern is typically present in only one or two very similar species, however, patterns 20 and 19 are present in six or more Bacteroidaceae species (Supplementary Dataset 1, tab 3). An analysis of the flanking DNA showed that they are contained on similar predicted MGEs of ~12–18 kb, that we designated MGE1 (pattern 20) and MGE2 (pattern 19) (Fig. 1b). MGE1 and MGE2 are nearly identical, with the area of divergence beginning within the gene upstream of the CDCL$^S$ gene and extending through the entire CDCL$^S$ gene (Fig. 1b). These MGE contain a site-specific recombinase that likely dictates excision and integration at the tRNA-Ser gene.

### Metagenomic analysis of CDCL genes in human gut microbiomes

Seven of the 28 CDCL genomic patterns are from species residing in the human gut. These CDCL-encoding genes were mapped to human gut metagenomic sets comprising 1958 metagenomes from 16 different studies (references in Supplementary Dataset 2, Tab3). In some datasets, a large percent of the samples contained at least one 2- or 3-component CDCL pattern (Fig. 1c, Supplementary Dataset 2). For example, in the Japanese dataset, 72% of the 614 samples contained these multi-component CDCL encoding genes. In other datasets, such as individuals from Madagascar, few of the metagenomic samples contained these CDCL genes, which may correlate with a low abundance of *Bacteroides/Phocaeicola/Parabacteroides* in these populations. The three CDCL group (pattern 1), confined to *P. vulgatus* and *P.*

*dorei*, are the most prevalent, detected in 511 of the 1958 metagenomes (26.1%). Patterns 14 and 26 were not detected in any metagenomic sample. MGE1 (pattern 20) and MGE2 (pattern 19) were detected in 254 and 172 metagenomes, respectively, and many metagenomes contained both MGEs. This high frequency is consistent with these loci transferring within co-resident strains in individual microbiomes. Collectively, these show that multi-component CDCL genes are widely distributed in the gut microbiota of geographically distant human populations.

### *E. anophelis* CDCL$^S$ component resembles complement MAC C9

We next sought to understand the structures of the 2-component CDCLs, especially that of the smaller component of the *E. anophelis* CDCL pair and of the Bacteroidaceae (represented by the *B. fragilis* MGE1 [pattern 20]) 2-component CDCLs termed BfCDCLs. We previously solved the CDCL$^L$ crystal structure from *E. anophelis* (EaCDCL$^L$, PDB: 8G33)[7], which resembled the 4-domain structure of the CDCs[1] (Fig. 2). Herein we solved the crystal structure of the smaller component of the *E. anophelis* CDCL pair (EaCDCL$^S$, PDB: 8G32), which lacks an analogous region to domain 4 but retains the general structure of domains 1–3 of the CDCs and CDCL$^L$. EaCDCL$^S$ also resembles the core structure of the complement C9 component which interacts with the C8α (Fig. 2) subunit of the complement C5b8 complex to assemble its β-barrel pore[9] (Fig. 2). A representative electron density map used to model the EaCDCL$^S$ is shown in Supplementary Fig. 1. The *B. fragilis* BfCDCL$^L$ and BfCDCL$^S$ structures were available in the AlphaFold database[10] (designations BF1276 and BF1275, respectively) and resemble their *E. anophelis* analogs (Fig. 2).

The domain 4 of the CDCs plays a role in cholesterol-binding whereas domain 3 contains the archetype protein fold that forms the membrane penetrating β-hairpins, the latter of which is also conserved in the MAC/perforin[6] and stonefish toxin[11] families of pore-forming proteins/toxins. The domain 3 α-helical bundles (Fig. 2, highlighted in red and yellow) refold into the membrane-spanning β-hairpins that, upon the oligomerization of ~36 monomers form the large CDC β-barrel pore[2,3,12]. Neither the primary (Supplementary Fig. 2) nor tertiary (Fig. 2) structures of the putative binding domains of the EaCDCL$^L$ and the BfCDCL$^L$ exhibit similarity to that of the CDC binding domain 4, and they lack any primary structural similarity with each other suggesting neither binds to cholesterol-rich membranes, and that the EaCDCL$^L$ and BfCDCL$^L$ each likely bind different receptors.

Notably, the CDCL$^L$ structures of both species lack the conserved β-tongue structure of the CDCs (shown in blue on PFO crystal in Fig. 2 and primary structures in supplementary Fig. 2), which is not resolved in the crystal structures of EaCDCL$^L$ or EaCDCL$^S$ (sequences shown in Supplementary Fig. 2). Although the AlphaFold models of the BfCDCLs show the analogous regions to be structured (Fig. 2, shown in blue), but not as a β-tongue, the predicted local distance difference test and predicted aligned error (Fig. 2 and Supplementary Fig. 3) suggests the predicted structure in this region is unreliable. The conserved β-tongue structure in the CDCs is thought to communicate conformational information from domain 4 upon membrane binding to induce conformational changes in domain 3 that allow monomers to interact and form the oligomeric pore[13]. As shown in the next section, the analogous regions in the CDCLs are also important for their activation to form a pore, but by a completely different mechanism.

### Pore formation requires proteolytic activation of CDCL$^L$ and CDCL$^S$

The low and inconsistent pore-forming activity previously noted by us[7] of the EaCDCL pair suggested an activation step was necessary, therefore, we tested the possibility that the CDCLs require proteolytic processing for activation. Trypsin was used to treat the EaCDCL and the BfCDCL pairs in the presence of 1-palmitoyl-2-oleoyl-glycero-3-phosphocholine (POPC) liposomes loaded with a carboxyfluorescein

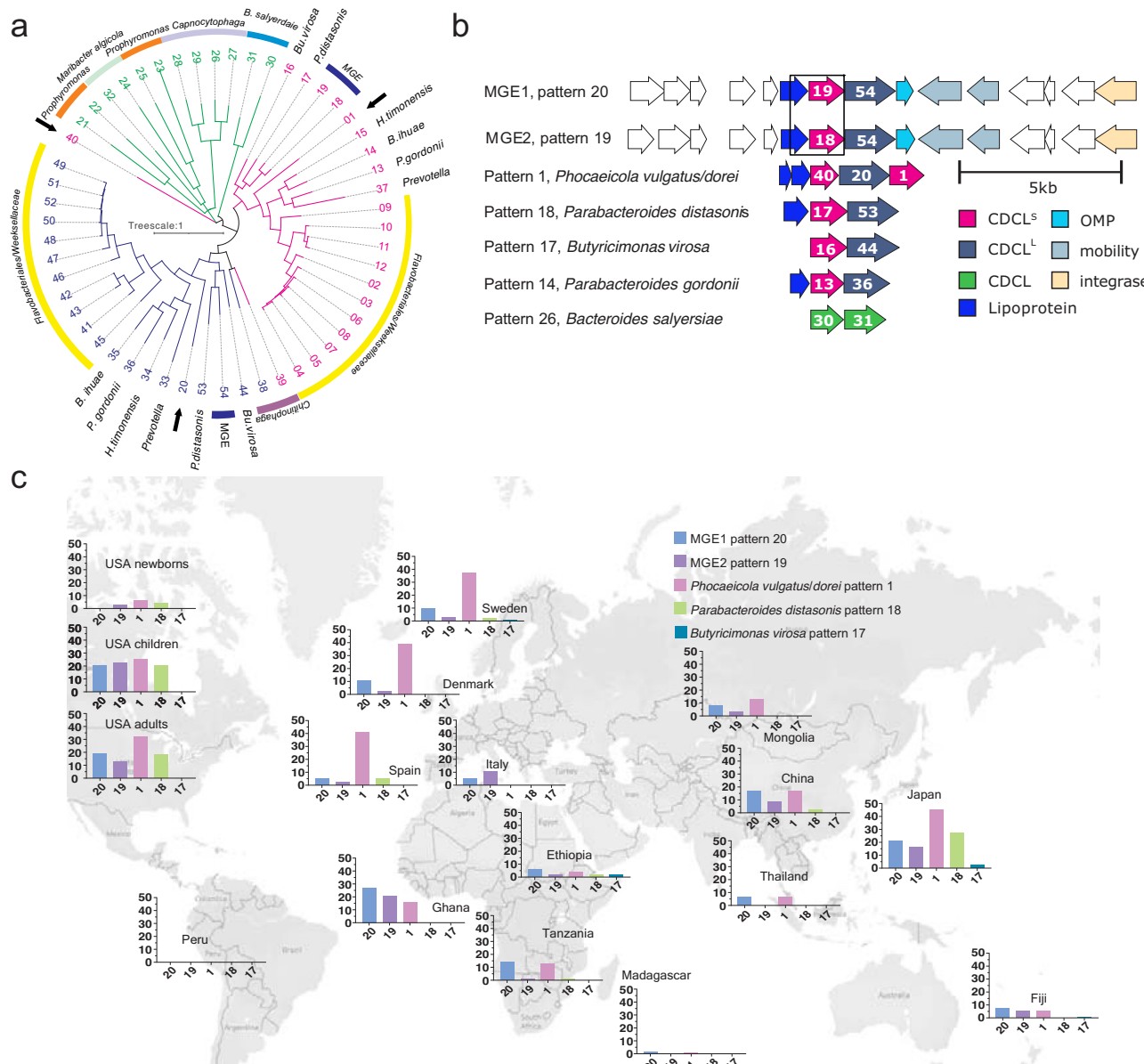

**Fig. 1 | Abundance and diversity of the Bacteroidota 2 and 3-component CDCL proteins. a** Phylogenetic tree of the 54 clusters of CDCL proteins encoded by pairs or triplets of adjacent genes. Pink designates proteins of the short subunit, CDCL$^S$, blue designates proteins of the long CDCL$^L$ subunit, and green designates proteins of pairs with similar-sized CDCLs. The numbers designate the cluster number (Dataset 1, tab1). **b** Orf maps of the seven CDCL gene pairs or triplet present in gut Bacteroidales genomes. Genes encoding CDCL$^S$ are colored pink, CDCL$^L$ is colored blue-gray, and CDCLs of similar-sized pairs are colored green. Genes that are predicted to be co-transcribed with the CDCL genes are included (white). For MGE1 and MGE2, the entire mobile genetic elements are shown, and the regions where the two MGE diverge are boxed. Predicted functions of some of the products of the genes on the MGEs are indicated. **c** Analysis of the prevalence of each of the seven CDCL gene pairs or triplets in human gut metagenomic datasets. The pattern 14 and pattern 26 CDCL gene pairs from *Parabacteroides gordonii* and *Bacteroides salyersiae*, respectively, were not detected above the designated cutoff threshold in any metagenome (Dataset 2) and are not included in the world map. Map data copyrighted OpenStreetMap contributors and available from https://www.openstreetmap.org. https://doi.org/10.1371/journal.pgen.1009541.g007.

(CF) marker. As pores form in the liposomes, the CF is released, which restores its fluorescence emission resulting in an increased emission intensity[14]. Trypsin treatment resulted in high and reproducible pore-forming activity for both the EaCDCL and BfCDCL pairs (Fig. 3a), which increased as the ratio of CDCL$^S$ to its cognate CDCL$^L$ increased and flattened at a ratio of 25–30 CDCL$^S$ to 1 CDCL$^L$ (Fig. 3a). Liposomes treated with unactivated CDCLs or activated CDCL$^L$ or CDCL$^S$ alone exhibited less than 10% of the maximal CF release achieved at the 30:1 molar ratio of activated CDCL$^S$ to CDCL$^L$. In a kinetic CF release assay, the injection of trypsin into a mixture of a 1:10 molar ratio of CDCL$^L$ to CDCL$^S$ results in the rapid activation of the CDCLs and release of the CF marker from the liposomes (Fig. 3a insert). Hence, the need for

proteolytic activation explains the absence of significant pore-forming activity we previously noted for the EaCDCLs[7].

Separation of the proteoliposome mixtures by SDS-agarose gel electrophoresis (SDS-AGE)[15] revealed that the BfCDCL$^S$ formed a relatively discrete oligomeric band with or without BfCDCL$^L$ (Supplementary Fig. 4). The EaCDCL$^S$ alone and combined with EaCDCL$^L$ appeared as a high mass smear, which may be due to a propensity of its oligomeric complexes to form SDS-resistant aggregates. The BfCDCL oligomer is smaller than that of PFO, which is likely due to the fewer CDCL$^S$ monomers (25–30) per oligomer than that observed for PFO oligomers (~36 monomers)[12]. The BfCDCL$^S$ alone formed the same size oligomers whether BfCDCL$^L$ was included or

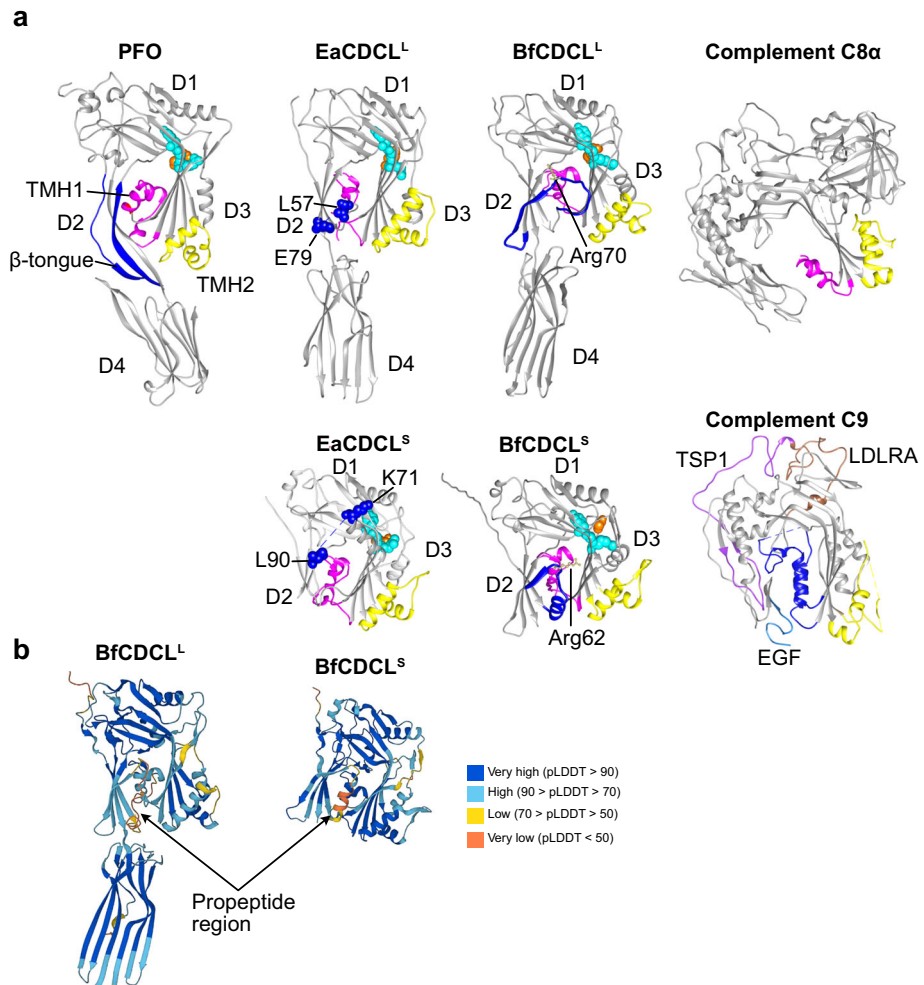

**Fig. 2 | CDC, CDCL, and complement C8α and C9 structures structures. a** Crystal structures of the archetype CDC perfringolysin (PFO) from *Clostridium perfringens*[1] (PDB: 1PFO), the EaCDCL[L] (PDB: 8G33) and EaCDCL[S] (PDB: 8G32) proteins from *E. anophelis*[7] and complement proteins C8α[47] (PDB: 2QQH) and C9[48] (PDB: 6CXO). The BfCDCL[L] and BfCDCL[S] structures were obtained from the AlphaFold2 database[10] (AlphaFold2 designations BF1276 and BF1275, respectively). The cognate α-helical bundles are shown that refold into the transmembrane β-hairpins 1 (TMH1, magenta) and 2 (TMH2, yellow), which form the β-barrel pore in all structures. The PFO β-tongue is shown in blue, whereas the analogous regions are not resolved in the crystal structures of EaCDCL[L (7)] and EaCDCL[S] (solved herein), but the bounding residues are shown as blue space-filled atoms. The AlphaFold2 models of BfCDCL[L] and BfCDCL[S] do not show their SpII signal peptides. Whereas the analogous regions to the CDC β-tongue are not visualized in the crystal structures of the EaCDCLs, the AlphaFold2 models of the BfCDCLs show the analogous regions as folded (ribbons colored in blue). However, the pLDDT confidence metric falls below 50, and the PAE metric is high in this region (Supplementary Fig. 3, and **b**, below), suggesting that the regions structures are unreliable. The YGR and GG motifs common to the CDCs and CDCLs[7] are shown in cyan and orange space-filled atoms, respectively. Arg70 and Arg62 are shown in tan atoms in BfCDCL[L] and BfCDCL[S], respectively. The GG motif largely lays behind the YGR motif atoms in the image. The core C9 structure resembles that of the CDCLs domains 1–3 (shown in gray). The C9 ancillary domains are shown in color (EGF, epidermal growth factor: blue, TSP1, thrombospondin type 1: purple and LDLRA, low-density lipoprotein receptor Type A: brown), which do not correspond to any structures in the CDCL[S]. **b** Shown are the pLDDT values colored onto the AlphaFold2 models of BfCDCL[L] and BfCDCL[S] from the AlphaFold2 structural database[10].

not, but in the absence of the CDCL[L] it does not efficiently form a pore (Fig. 3a).

## Identification of protease cleavage sites in EaCDCLs and BfCDCLs

Bacteroidales species produce a family of C11-type proteases that are synthesized as lipoproteins allowing them to be anchored to the outer membrane and exposed on the cell surface[16]. We determined that trypsin, and C11-type proteases from *B. fragilis* (fragipain) and *Phocaeicola dorei* (DpnB[17]), cleaved BfCDCL[L] after TR[70] and BfCDCL[S] after TR[62], whereas EaCDCL[L] was cleaved after TK[66] and EaCDCL[S] after TK[87] (Supplementary Fig. 5). For the EaCDCLs, proteinase K also cleaved EaCDCL[L] after TK[66], whereas it cleaved EaCDCL[S] just upstream of its TK[87] site after IA[78]. A second C11-type surface protease from *P. dorei*, DpnA[17], did not cleave either protein. The in vitro activation site for

each CDCL pair resides in their regions that are analogous to the β-tongue region of the CDCs.

## Interaction of CDCL[L] and CDCL[S]

The endpoint analyses predict that CDCL[L] only interacts with CDCL[S] but that CDCL[S] also interacts with itself to form the oligomeric pore (Fig. 3a). Förster resonance energy transfer (FRET) experiments were used first to examine the interaction of the two subunits for the EaCDCL and BfCDCL pairs. A single engineered cysteine was introduced into each protein, and the sulfhydryl was labeled with a maleimide derivative of either Alexa-488 (donor fluorophore or D) or tetramethylrhodamine (acceptor fluorophore or A). Donor emission is quenched when the D- and A-labeled proteins form a stable interaction that brings the D and A probes within the Förster distance (~5 nm for this fluorophore pair[18]).

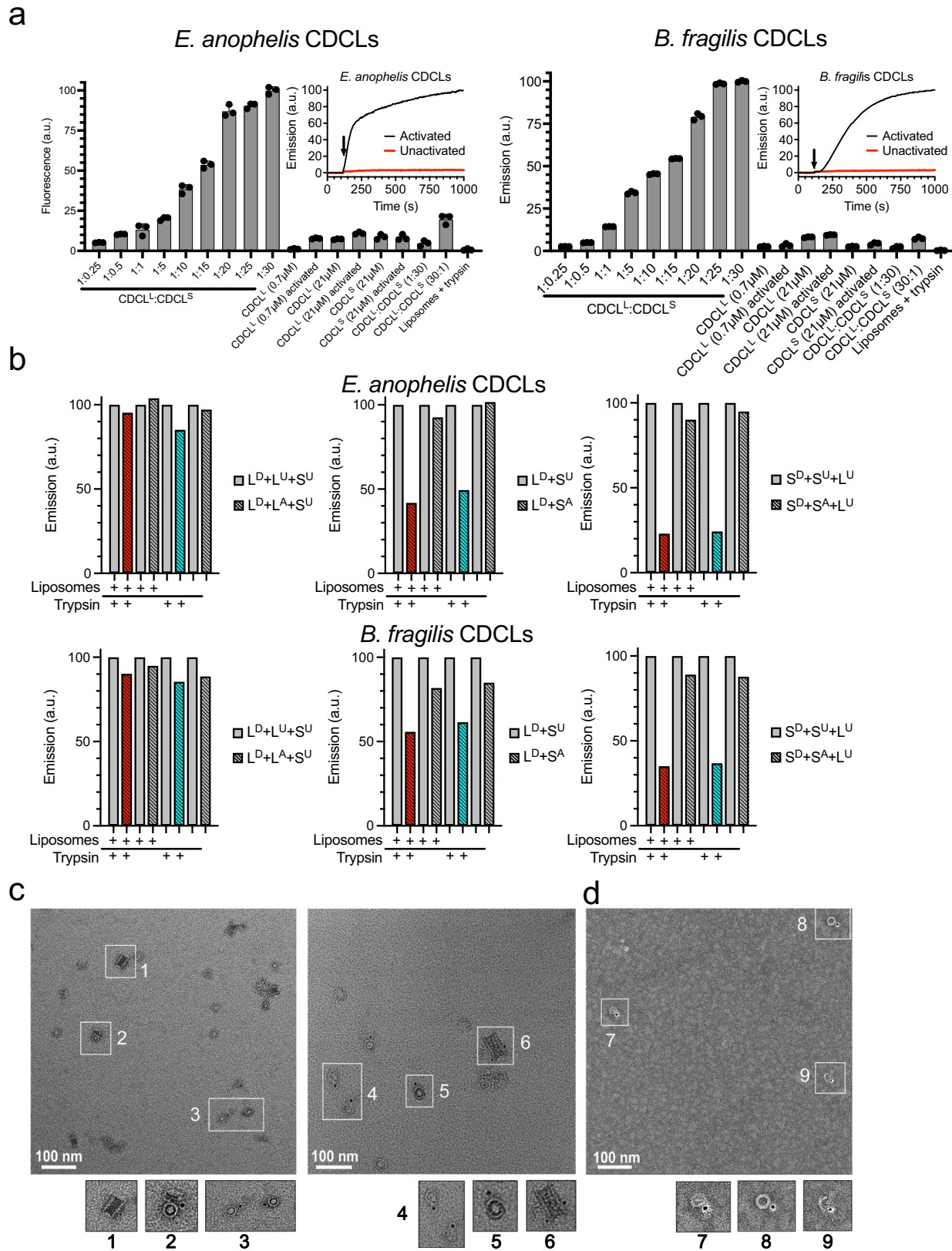

Little quenching of proteolytically activated D-labeled EaCDCL^L or BfCDCL^L was observed in the presence of A-labeled CDCL^L (red and cyan color bars, Fig. 3b, left bar plots) in the presence of unlabeled CDCL^S. However, significant quenching of the proteolytically activated D-labeled CDCL^L or D-labeled CDCL^S was observed when either was mixed with A-labeled CDCL^S, whether or not liposomes are present

(red and cyan color bars in Fig. 3b, center and right bar charts). These results show that CDCL^L does not interact with itself, but CDCL^S interacts with CDCL^L and with itself. Also, unactivated CDCLs do not interact, as little quenching is observed for unactivated D-labeled CDCL^L or CDCL^S in the presence of A-labeled CDCL^S. Some quenching is observed for the unactivated D-labeled BfCDCL^L, as we could not

**Fig. 3 | CDCL pore-forming activity and oligomer assembly. a** Endpoint analysis shown in the first 9 bars represent various ratios of trypsin-activated CDCL$^L$ (0.7 μM) to CDCL$^S$ from *E. anophelis* AG1 or *B. fragilis* YCH46 in the presence of carboxyfluorescein (CF)-loaded phosphatidylcholine (POPC) liposomes. After 30 minutes the fluorescent intensity of the CF was determined. In the last 8 bars, various controls were run as denoted in the *X* axis labels. (**a**, inset) The kinetics of CF release by EaCDCL and BfCDCL from POPC liposomes. Trypsin was injected at 120 seconds (arrow) to initiate activation of CDCLs. Endpoint experiments were performed in triplicate and the kinetic data in the insets is representative of 2 or more individual experiments. **b** For the Förster resonance energy transfer (FRET) analyses the total CDCL$^L$ and CDCL$^S$ are maintained at a 1:10 molar ratio. L$^D$ and S$^D$, donor-labeled CDCL$^L$ and CDCL$^S$, respectively. L$^A$ and S$^A$, acceptor-labeled CDCL$^L$ and CDCL$^S$, respectively. L$^U$ and S$^U$, unlabeled CDCL$^L$ and CDCL$^S$, respectively. The emission of the D + U sample was set to 100 arbitrary units (a.u.) and D + A sample was divided by the emission of the D + U sample to provide the extent of quenching by the presence of the acceptor-labeled protein, as done previously[49]. The red and cyan highlight the D + A samples show the addition of trypsin and liposomes (red) or trypsin alone (cyan). Each pair of bars in the spectroscopic data is representative of 2 or more individual experiments. **c** Representative micrographs from grids prepared from a sample of activated-EaCDCL$^S$ (0.3 μM) + EaCDCL$^L$ (0.15 μM) and incubated on POPC lipid monolayers at 37 °C for 25 minutes prior to staining. Only the EaCDCL$^L$ was labeled with a 5 nm nanogold probe at N239C. **d** Representative micrograph from grids prepared from a sample using equimolar concentrations of activated-EaCDCL$^S$ (0.4 μM) + EaCDCL$^L$ (0.4 μM) on lipid monolayers. Not all oligomers show a gold-labeled EaCDCL$^L$ since EaCDCL$^S$ can oligomerize in the absence of EaCDCL$^L$. Controls where the gold-labeled EaCDCL$^L$ and EaCDCL$^S$ are incubated alone are shown in Supplementary Fig. 6.

prevent a fraction of the BfCDCLs from being activated during its expression in *E. coli* and/or purification.

The results of the FRET analysis were confirmed by demonstrating that a single nanogold labeled EaCDCL$^L$ is present in each oligomeric complex by transmission electron microscopy (Fig. 3c), even when equimolar amounts of EaCDCL$^L$ and EaCDCL$^S$ are used (Fig. 3d). Measurements of oligomeric ring complexes provide an estimated inner diameter of ~20 nm that yields an inner circumference of ~63 nm. When the α-helical bundles of the CDCs and CDCLs are extended into the twin transmembrane β-hairpins, their combined width is ~2.1 nm, which results in an estimated 30 monomers per oligomeric complex, consistent with the endpoint analysis. These studies strongly suggested that the CDCL$^L$ acts as the membrane platform to recruit many CDCL$^S$ to form the membrane pore, which is explored next.

### CDCL$^L$ is the membrane platform that recruits CDCL$^S$ to assemble the pore

The receptors for the EaCDCL$^L$ and BfCDCL$^L$ proteins are unknown, however, the above studies show that a single CDCL$^L$ is required for pore formation, presumably by acting as the membrane anchor from which a pore assembles by recruiting ~30 CDCL$^S$. To directly show that CDCL$^L$ functions as the membrane platform, the putative EaCDCL$^L$ binding domain (residues 369–516) was substituted with the cholesterol-binding domain of PFO (residues 389–500, domain 4 in Fig. 2), which has no primary or structural similarity with the analogous region of EaCDCL$^L$ (Fig. 2 and Fig. S1) and uses cholesterol as its receptor[19].

When the EaCDCL$^L$-PFO(D4) chimera is combined with native EaCDCL$^S$, it induced the release of CF marker from POPC-cholesterol liposomes but not POPC liposomes (Supplementary Fig. 7a, left panel) upon proteolytic activation. Proteolytic activation of the chimera is still required, as the unactivated proteins do not form pores in the POPC-cholesterol liposomes (Supplementary Fig. 7a, right bottom panel). These studies show that even though the EaCDCL$^L$ is redirected to a different receptor, it retained the ability to recruit the native EaCDCL$^S$ to form a pore. We also tested the native and EaCDCL$^L$-PFO(D4) chimera on sheep erythrocytes and showed that the native toxin did not lyse the cells, but if the native EaCDCL$^S$ was mixed with the EaCDCL$^L$-PFO(D4) chimera, lysis was observed (Supplementary Fig. 7b). The highly variable nature of the putative binding domain of the CDCLs and their ubiquitous presence in many gut microbes suggested they are not eukaryotic toxins, as are the CDCs, but antibacterial toxins. Since antibacterial toxins of gut microbes typically target closely related species[17,20], we next tested the susceptibility of various *Bacteroides* and *Phocaeicola* species/strains to the BfCDCL pair.

### Biological activity of the CDCLs

We investigated whether the CDCLs target other bacteria using the BfCDCL pair, as *B. fragilis* is genetically tractable, and this pair is encoded on a MGE and widely distributed in gut Bacteroidaceae. Protease-activated and unactivated BfCDCLs had no or minimal effects on the growth of the BfCDCL-producing strain *B. fragilis* YCH46 or several other *Bacteroides* strains (Fig. 4a). However, both protease-activated and unactivated BfCDCLs strongly inhibited the growth of *Phocaeicola dorei*, *Phocaeicola vulgatus*, and *Bacteroides uniformis* strains. Importantly, little difference was observed between the antibacterial activity of activated and unactivated BfCDCL proteins, suggesting both proteins are activated by protease(s) on the surface of sensitive strains. Controls with protease alone, protease-activated BfCDCL$^L$ or BfCDCL$^S$ individually or 10 times of the amount of CDCL$^L$ alone did not affect the growth of the sensitive *P. dorei* 9_1_42FAA strain (Fig. 4a, bottom row). We next examined the effect of the BfCDCLs on the *P. dorei* 9_1_42FAA membrane integrity.

### The BfCDCLs disrupt the outer and inner membranes of bacterial cells

Since the CDCLs form large pores in liposomes (Fig. 3c, d), they should cause leakage of the periplasmic proteins from BfCDCL-treated bacteria. A culture of *P. dorei* 9_1_42FAA was treated at mid-log growth with the BfCDCL pair, which resulted in subsequent growth inhibition (Supplemental Fig. 8a). When the culture media from BfCDCL-treated and untreated cells was subjected to SDS-PAGE, many bands in the 25–125 kDa range are present in the culture supernatant from BfCDCL-treated culture, which are not apparent in the supernatants of untreated cells (Supplementary Fig. 8b, c). Mass spectrometry of selected bands excised from the gel yielded mostly periplasmic proteins (Supplementary Dataset 3) of up to ~125 kDa, as well as stress response proteins DnaK and the OmpH/Skp periplasmic chaperone, which demonstrates that BfCDCL pores allow small and large proteins to leak out of the periplasm.

The integrity of the inner membrane was also analyzed using propidium iodide (PI), as it only fluoresces upon binding to DNA. The fraction of cells that fluoresce increased over time for the BfCDCL-treated samples, whereas the untreated cells did not show an increase over the same time (Fig. 4b, c). Hence, BfCDCL kills cells by opening a large pore in the outer membrane, leading to the loss of the periplasmic contents and inner membrane integrity.

### In vivo activation of the BfCDCLs and identification of the activating protease

The fact that the BfCDCLs are activated by the sensitive strains suggested that the activating protease is present on their surface. We first determined whether the protease cleavage sites identified in vitro (Supplementary Fig. 5) are the same sites required for activation in vivo. The BfCDCL$^L$ and BfCDCL$^S$ cleavage site mutants (R$_{70}$A and R$_{62}$A, respectively) were combined with their cognate wildtype protein and used to treat *P. dorei* 9_1_42FAA (Fig. 5a). In both cases the combinations were largely inactive (Fig. 5a), which demonstrates the in vitro activation sites are the primary activation sites in vivo. This was also

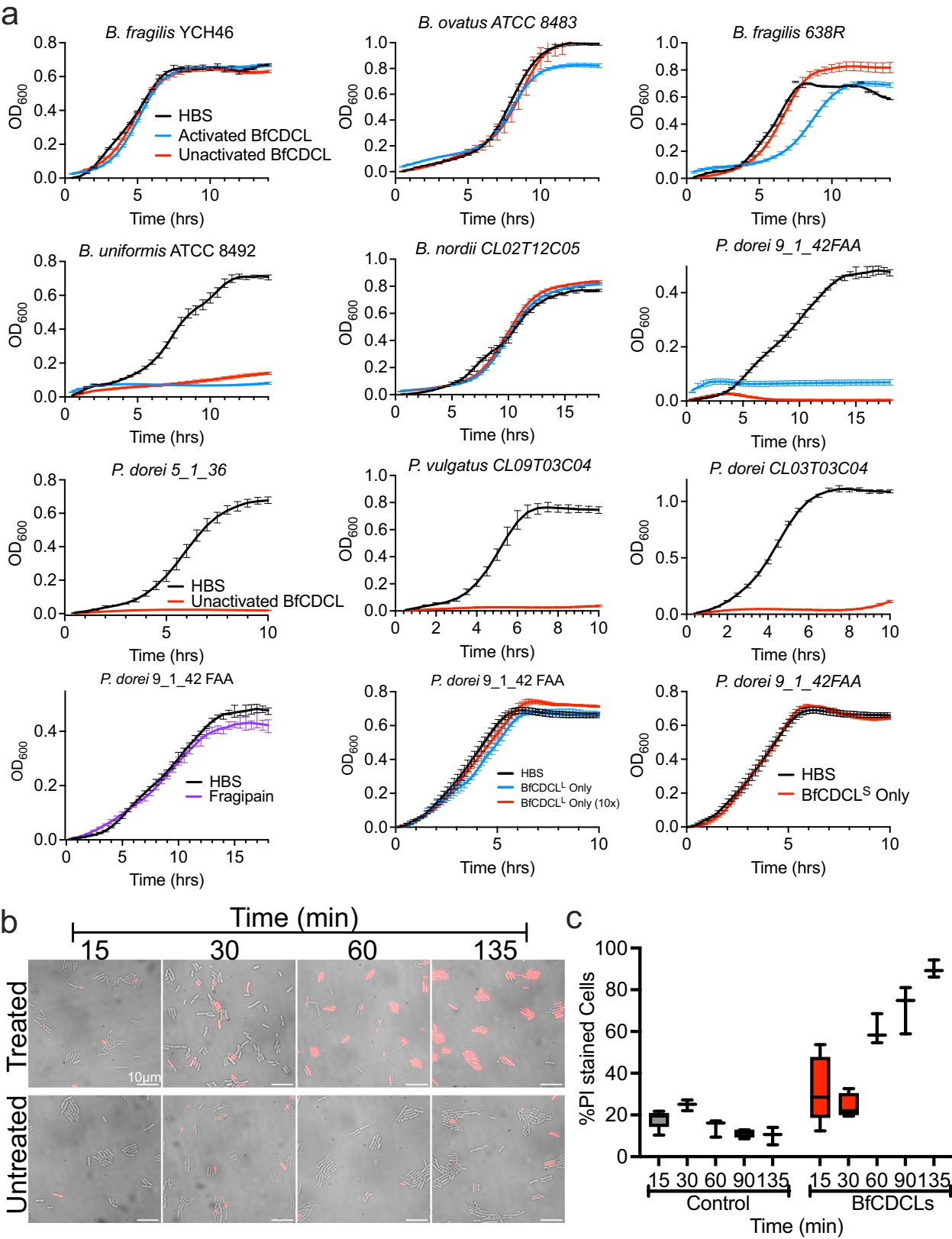

consistent with in vitro cleavage of BfCDCL$^L$ and BfCDCL$^S$ wildtype, and the R$_{70}$A and R$_{62}$A mutants, with purified DpnB (Supplementary Fig. 9).

Next, we identified the activating protease of a sensitive species. The BfCDCL-sensitive *P. dorei* and *P. vulgatus* strains carry genes for the surface C11-type proteases, DpnA and DpnB, which we previously identified and individually deleted from *P. dorei* CL02T00C15[17]. The wildtype *P. dorei* CL02T00C15 strain is sensitive to the unactivated BfCDCLs (Fig. 5b). Deletion of *dpnB*, but not *dpnA*, renders this bacterium resistant to the BfCDCLs (Fig. 5b), indicating that DpnB is the proteolytic activator, which is consistent with our in vitro findings that DpnB but not DpnA cleaves the CDCLs.

**Fig. 4 | BfCDCLs target *Bacteroides* and *Phocaeicola* species and disrupt inner and outer membranes. a** Several *Bacteroides* and *Phocaeicola* species/strains at an $OD_{600}$ of ~0.6 were diluted 1:30 into fresh media, and equal amounts of unactivated or fragipain-activated BfCDCL$^L$ + BfCDCL$^S$ (at a 1:10 molar ratio) or HBS were added to the culture (150 μL total volume in a 96-well microtiter plate). In the third row the strains were only treated with unactivated BfCDCLs. In the first panel of the fourth row, *P. dorei* CL03 was treated with HBS or fragipain. Note that the assays for *P. dorei* in **a** (second row, last panel and fourth row, first panel) used the same HBS control, as they were performed simultaneously. In the second and third panels in the fourth row *P. dorei* 9_1_42FAA was treated with unactivated BfCDCL$^L$, a 10x BfCDCL$^L$ or BfCDCL$^S$ alone. Both panels used the same HBS treated cell control, as they were also run simultaneously. The standard error of the mean of the $OD_{600}$, derived from three individual growth assays, is shown for each datapoint. **b** Time-course microscopy analysis and quantification of membrane disruption of *P. dorei*

9_1_42FAA cells treated with toxin or HBS as a negative control. The toxin (or HBS) was added to exponentially growing cells, and at each time point cells were imaged in **b** by differential interference contrast (DIC) and epi-fluorescence microscopy in the presence of propidium iodide (PI), a red fluorescent indicator of compromised cell membranes by accessing and intercalating into DNA. The percentage of dead cells was quantified in **c** up to the 135-minute mark. For quantification of propidium iodide-stained cells, at least 550 cells in three to five separate fields of view were counted for each time point. The data are presented as a standard box plot wherein the boxes represent the quartile distribution of the PI-stained cells, and the horizontal line represents the data median. The whiskers represent the highest and lowest number of PI stain cells from all 5 views. All bacterial growth curves represent the mean of 3 individual cultures (except *B. uniformis* treated with unactivated BfCDCL, which was done in duplicate) with the standard error of the mean of the $OD_{600}$ for each time point.

## Inhibition of BfCDCL-mediated pore formation by an immunity protein

The operon containing the CDCL genes of the BfCDCL region (MGE1, Fig. 1) contains two additional genes: a gene immediately upstream of CDCL$^S$ that encodes a 30.4 kDa predicted outer surface localized lipoprotein and a downstream porin (Fig. 1b). A recombinant version of the lipoprotein was generated where the SpII signal peptide was replaced by a His$_6$ tag and purified from *E. coli* as a soluble protein. The recombinant protein effectively inhibits BfCDCL-mediated release of CF marker from liposomes at a lipoprotein/BfCDCL$^S$ ratio as low as 3:10 and significant inhibition is also observed at a 1:10 molar ratio (Fig. 5c). Pore-forming activity is also rapidly abrogated when the recombinant protein is added after pore formation is initiated by protease addition (Fig. 5c, second row, last panel). Furthermore, when the gene encoding this lipoprotein is deleted in the BfCDCL-producing strain *B. fragilis* YCH46, it becomes more sensitive to its own CDCL pair (Fig. 5d). Both the activated and unactivated BfCDCL exhibit the same effect when the the *B. fragilis* YCH46 lipoprotein is not present, which suggests its own CDCLs can be activated by a protease on its surface. Next, we cloned and expressed it in sensitive cells and found that they become largely resistant to the activated and unactivated BfCDCL pairs (Fig. 5d). Hence, we named this protein BcdI, for Bacteroidales CDCL immunity protein. BcdI is likely indispensable for host cell protection when the MGE carrying these toxin genes is transferred into sensitive species/strains.

## Discussion

The studies herein reveal for the first time a distant evolutionary branch of the CDCs is present in many bacterial species that include important species of the human gut Bacteroidota, which are widespread in geographically dispersed human populations. They utilize a 2-component assembly mechanism not previously observed for any other pore-forming toxin (shown schematically in Fig. 6a) but exhibit a striking similarity to the mammalian complement MAC pore assembly mechanism. A deeper analysis of the *B. fragilis* 2-component CDCL activity revealed that, unlike the CDCs, it targets bacterial rather than eukaryotic cells. This may also be true for the EaCDCLs, as *E. anophelis* is a commensal in the malarial mosquito midgut[21]. The capacity of the BfCDCLs to kill other related bacterial strains, and the presence of distinct evolutionary variations of the CDCLs, suggest that they play important roles in species/strain competition in the diverse environments of these Bacteroidota species.

The receptor for EaCDCL$^L$ or BfCDCL$^L$ is currently unknown; however, the CDCL$^L$ is the essential membrane-binding component, which recruits many CDCL$^S$ to form the β-barrel pore. This type of mechanism is only used by the complement MAC wherein a single C5b8 membrane complex recruits many C9 components to form its β-barrel pore. *B. fragilis* also protects itself from its own CDCLs with the BcdI immunity protein that inhibits the formation

of the CDCL pore, much like the mammalian GPI-anchored CD59[22], which protects host cells by inhibiting MAC pore formation. However, the loss of the BcdI from *B. fragilis* only exhibits a partial loss of protection, suggesting there may be other factors that help protect it from its own toxin (Fig. 5d, left panel). Consistent with this observation, substantial but not complete protection was seen when the BcdI was introduced into sensitive strains (Fig. 5d, right 2 panels). This is not unexpected, as there may be other reasons *B. fragilis* was not completely sensitive to its own toxin, which remain to be revealed once the system is better understood. This is also true for the complement MAC, as multiple factors are also required for host protection from the MAC[23]. Not all 2-component systems contain genes encoding obvious immunity proteins, as the *E. anophelis* CDCL operon does not contain other co-transcribed genes. The absence of an immunity protein suggests that some CDCL-producing bacteria may not be susceptible to their own CDCLs.

BfCDCL killing of sensitive strains requires the C11 surface protease, DpnB, which was shown to cleave and activate both CDCL components in vitro and is required for in vivo activation on sensitive bacteria. As shown in Fig. 6b, cleavage of the CDCL$^L$ propeptide likely allows β5 (Fig. 6b, blue strand) to be displaced from its interaction with β4 (Fig. 6b, red strand), which is the interface that interacts with β1 (Fig. 6b, cyan strand) of the first incoming monomer of CDCL$^S$. The propeptide of the CDCL$^S$ blocks β1, which must be exposed to interact with β4 of CDCL$^L$ and of other CDCL$^S$ that oligomerize to form the β-barrel pore. Bacteroidaceae produce two antibacterial toxins whose secretion, and/or activity, is dependent on the surface C11 protease of the producing cell, which activates then during secretion[17]. The proteolytic activation of an antibacterial toxin on the target cell, like that observed herein for the BfCDCLs, has not been reported for any other antibacterial toxin (that we know of); this is a characteristic that is largely restricted to bacterial toxins that attack eukaryotic cells[24]. Activation on the target cell may increase their effectiveness, as we have observed that the CDCL$^S$ oligomerizes after proteolytic activation, which may remove it from the pool of monomers that can interact with membrane-bound CDCL$^L$ to form a functional pore. Interestingly, it was recently shown that effective killing of Gram-negative bacteria by the complement MAC requires that the C5 convertase first binds the bacterial surface where it cleaves C5 to initiate the assembly of the C5b8 membrane complex[25].

In summary, we have shown that the CDCLs employ a complement MAC-like mechanism of pore assembly to attack other bacteria, which are widespread in diverse species of the human microbiome. As many bacterially produced antibacterial toxins target strains of the same or closely related species[20], it will be interesting to study the range of targeting of the other pairs of CDCL toxins and their ecological effects in the gut microbiota.

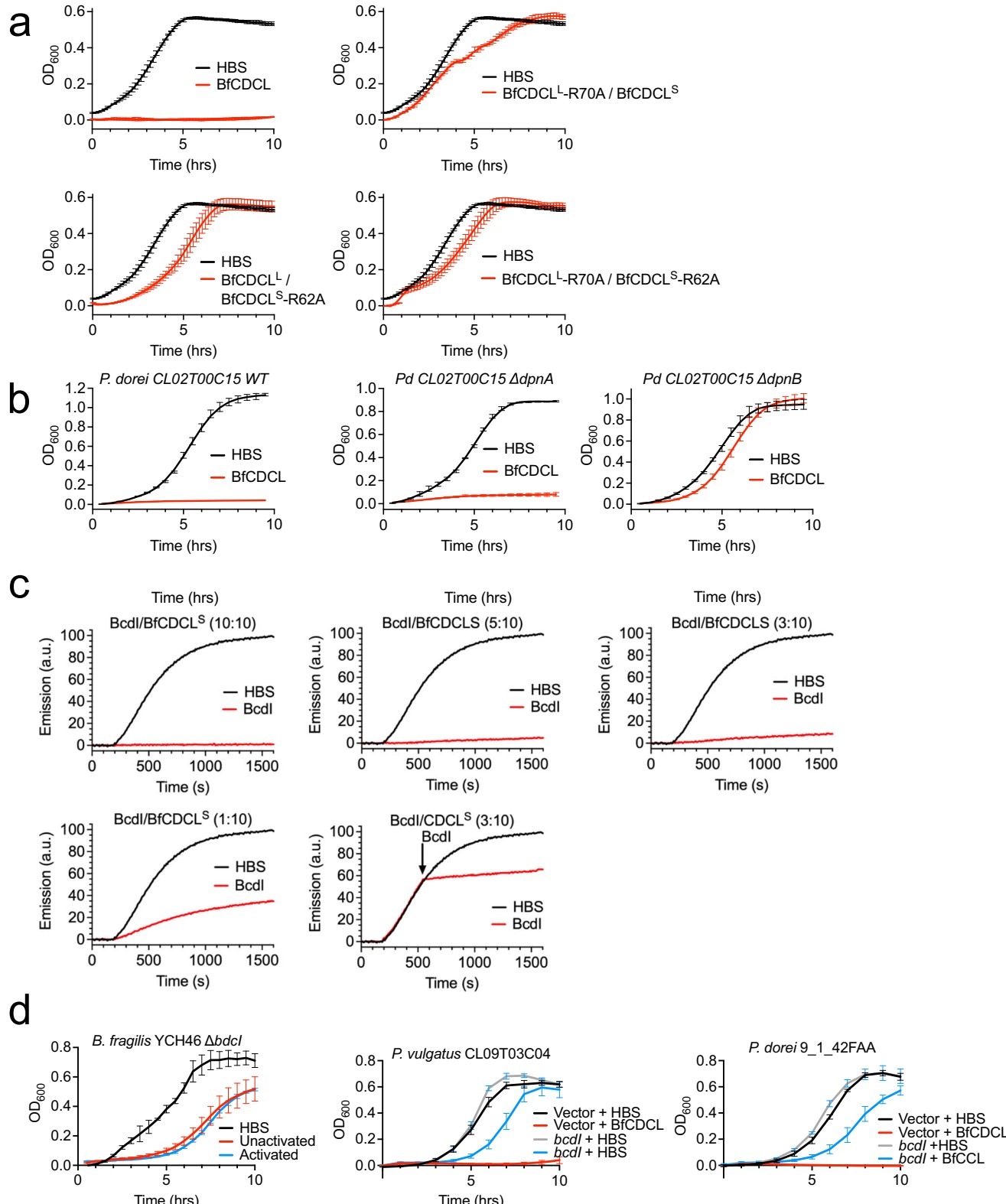

## Methods

### Bacterial strains, plasmids, and protein expression constructs

All bacterial strains used and created in this study are listed in Supplementary Table 3. The genes for recombinant CDCL genes and their derivatives, and for the C11-type proteases fragipain and DpnB were synthesized, and codon optimized for expression (Genscript) (Supplementary Table 3). The synthesized genes started after the cysteine codon of the SpII signal peptide and cloned into pET-15b between the *Nde*I and *Bam*HI restriction sites that placed a poly-histidine (His$_6$) tag and thrombin cleavage site at the amino terminus of each protein for purification.

### Identification and analysis of CDCL proteins

Our genome collection consists of 5947 RefSeq genomes identified by NCBI as belonging to the phylum Bacteroidota (synonym Bacteroidetes) excluding genomes flagged as anomalous or as being of

**Fig. 5 | BfCDCL proteolytic activation and protection by an immunity protein.**
**a** The proteolytic cleavage sites identified by the in vitro assays were tested for activity against the sensitive *P. dorei* 9_1_42FAA strain. The growth of *P. dorei* 9_1_42FAA in liquid cultures is shown when treated with unactivated wildtype BfCDCLs (as in Fig. 4) and the BfCDCL^L and BfCDCL^S proteolytic cleavage site mutants, R70A and R62A, respectively when paired with its wildtype cognate CDCL^L or CDCL^S. Note that the HBS control is the same in **a**, as they were performed simultaneously. **b** *P. dorei* CL02 and its DpnA and DpnB gene knockouts[17] were treated with the unactivated wildtype BfCDCLs. **c** Mixtures were prepared of unactivated BfCDCLs, CF liposomes and BcdI, which was added at 1:10, 3:10, 5:10, and 10:10 molar ratios to the BfCDCL^S. Fragipain (FP) (3.1 µg) was injected at 120 seconds to activate the BfCDCLs and pore formation was followed by the

release of CF. In the last panel of (**c**) the BcdI was injected at a 3:10 molar ratio to BfCDCL^S at 540 sec. The HBS control is the same for all these experiments, as they were performed simultaneously. **d** The BcdI gene was deleted from in *B. fragilis* YCH46 and added in trans to the sensitive *P. dorei* strains CL09T03C04 and 9_1_42FAA. Their sensitivity to unactivated BfCDCL pair was tested in the liquid culture assay (as in Fig. 4b). The *B. fragilis* YCH46 was also treated with fragipain-activated BfCDCLs. The growth curves in **a**, **b** represent the standard error of the mean of the $OD_{600}$ for each time point derived from three individual growth assays. The growth assays in **a**, **b** were performed on a Stratus Kinetic plate reader (Cerillo), whereas those in (**d**) were performed on a Biotek Epoch 2 plate reader. The CF release assays in **c** are representative of 2 assays.

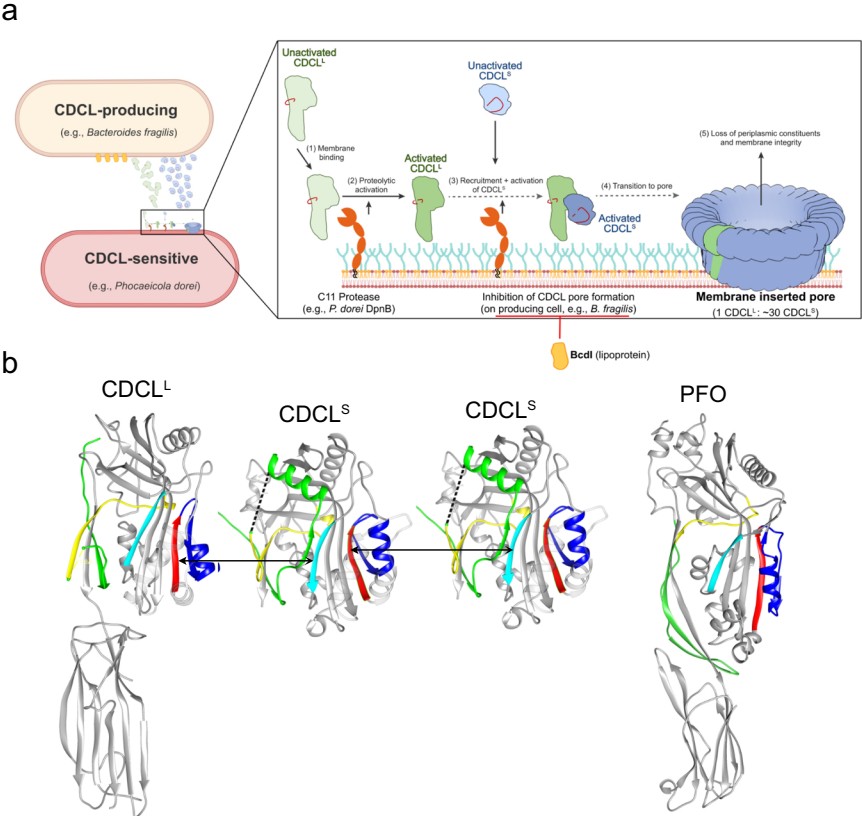

**Fig. 6 | Molecular mechanism model of the CDCL pore formation. a** The first step in the assembly of the pore involves the binding of the CDCL^L to its receptor and its activation by C11-type DpnB protease (or a protease activated by DpnB). Next, the CDCL^S is recruited to the bound CDCL^L and activated by the same protease as CDCL^L. Although the CDCL^S must be activated by DpnB it is not yet known how it is activated and recruited to the membrane-bound CDCL^L. The CDCL^S propeptide must be removed to allow its interaction with CDCL^L, as shown in **b**. More CDCL^S monomers are then recruited to the growing oligomer. Whether the CDCL^S monomers insert their transmembrane β-hairpins into the membrane successively as each is added to form a growing pore, which is suggested for complement MAC[48], or insert upon prepore completion, as suggested for the CDCs[15,50], remains to be determined. **b** Shown are the crystal structures of EaCDCL^L, EaCDCL^S, and

PFO. Upon proteolytic activation of CDCL^L, the propeptide (blue and dashed line) may be lost, but like the β-tongue of PFO[13,51] (blue), this action may also be conformationally coupled to weaken the interaction of β5 (blue) with β4 (red), as we have found in PFO[13,51]. Next, the CDCL^S propeptide must be cleaved and lost, or displaced, so that its β1 (cyan) can form backbone hydrogen bonds with β4 of CDCL^L, and as more CDCL^S enters the assemble, they must also form intermolecular interactions between β4 and β1. The same interactions take place between PFO monomers, but displacement of β5 from β4 is triggered by membrane binding, which is conformationally coupled via the β-tongue (blue)[13,51]. **a** created with BioRender.com, released under a Creative Commons Attribution-NonCommercial-NoDerivs 4.0 International license.

questionable taxonomy. This collection includes genomes from 8 Orders, 50 Families, 327 genera, and represents 1196 species plus 307 genomes identified to the genus level only. The 22,114,981 proteins included in this set were scanned for cholesterol-dependent cytolysin-like (CDCL) proteins with the Pfam Thiol-activated cytolysin HMM (PF01289.22) using hmmsearch from the HMMER[26] package (v. 3.3) with the gathering threshold bit score (25) set as a cutoff. This analysis identified 2468 proteins, 21 of which were partial proteins and were

eliminated, thus leaving 2447 full-length CDCLs for further analysis. These proteins were derived from 1578 genomes, with as many as eight proteins per genome (1 genome) to as few as one (821 genomes), while 736 genomes had two (675) or three (61) CDCLs.

This collection of 1578 genomes was further parsed to identify genomes containing two CDCL proteins that reside on the same contig, were encoded by the same strand, and that were near each other. This process retained 1098 proteins from 537 genomes.

An analysis of this set of proteins revealed a reasonably continuous size distribution range—the smallest was 117 amino acids, and the largest was 674 amino acids, with clustering observed at ~350, ~450, and ~550 amino acids. These CDCL proteins are all expected to contain N-terminal signal peptides (SpII). To avoid noise potentially introduced into the dataset by the gene calling algorithm selecting the wrong start codon, this set of 1098 proteins was scanned for SpII signal sequences using LipoP[27] (v. 1.0a). This analysis indicated that 845 of these proteins contained a recognizable SpII signal sequence, 25 were identified as SpI proteins, and 228 were identified as having no signal sequence.

Analysis of the returns from LipoP revealed that many of the non-SpII proteins had an SpII call just under the threshold used by LipoP for such a call. The non-SpII proteins were thus further scanned by a Perl regular expression for the peptide sequence [SA]C appearing in the first 35 amino acids. This analysis "rescued" 221 additional SpII proteins; the remaining 21 non-SpII proteins were removed from the dataset, leaving 1066 proteins from 537 genomes for further analysis.

Three segments were removed from consideration as the remaining CDCL protein was orphaned (no longer paired) after the SpII operation above. An additional 21 genomes were removed due to various defects involving the CDCL proteins (protein began or ended at the edge of a contig, ORF interrupted by an insertion element, ORF frameshifted, truncated, etc.). The remaining 1120 CDCL proteins from 497 genomes were clustered using Clustal[28] at 96% protein-level ID. (Supplementary Dataset 1, Tab 1).

A representative of each of the 28 CDCL pattern groups detected in the 497 genomes analyzed was selected (Supplementary Dataset 1, Tab 2), and these representative loci along with some flanking DNA was retrieved, and ORF maps of these regions were produced (Fig. S1).

## Metagenomic analyses

Seven of the 28 CDCL pattern group representative loci were from species residing in the human gut. *Bacteroides ihuae* Marseille-P2824 was isolated from the human respiratory microbiome and thus not included. The DNA spanning the CDCL proteins was used in metagenomic mapping analyses. We utilized a set comprising 1958 metagenomes from 16 different studies downloaded from the European Nucleotide Archive and mapped the reads to each of the seven representative loci of human microbiome origin. This set of microbiomes encompasses groups of individuals from various ethnic, cultural, gender, and lifestyle groups, as well as groups varying by age and health status. The read mapping was performed with use of BBsplit, a tool included in the BBMap suite of programs[29] (v. 38.90). This tool maps reads to multiple reference sequences simultaneously and will determine the best match in the case of ambiguity (the read maps to more than one reference) and count that read only once. To be considered present in a metagenomic sample, the reference sequence must have been covered at least 75% by a minimum of 20 reads and have reached an average of five-fold coverage per reference base.

## Expression and purification of recombinant proteins

The recombinant proteins used herein were purified as follows (except EaCDCL$^S$ used for crystallization and TEM). The CDCL genes were expressed in *Escherichia coli* BL21/DE3 pLysS and affinity purified[7]. Briefly, a 100 ml overnight culture of *E. coli* containing the expression plasmid was added to 1.7 L of fresh Terrific Broth (EMD Millipore) and grown to an OD$_{600}$ of 0.8, made 1 mM in IPTG (isopropyl β-D-1-thiogalactopyranoside, Gold Bio) and grown overnight at 18 °C. Cells were harvested by centrifugation, suspended in 50 ml Nickel A buffer (10 mM MES, 150 mM NaCl, pH6.5), and lysed using an Avestin EmulsiFlex-C3 cell disruptor. Cell debris was removed by centrifugation at 21,000 × *g*, and the supernatant was loaded onto and recirculated on a cobalt-loaded metal chelate column (Cytiva) for 60 min. The His$_6$ tagged recombinant protein was eluted using an AKTA Prime

Plus (Cytiva) to generate a linear 100 ml gradient from 0%–50% Nickel B buffer (10 mM MES, 150 mM NaCl, 1 M imidazole, pH6.5). Purified proteins were exchanged into HEPES buffered saline (HBS, 50 mM HEPES, 100 mM NaCl, pH 7.4) using a HiPrep 26/10 column (Cytiva). For the proteases, an additional purification step was included wherein the protein was passed over a Superdex 75 10/300 (Cytiva) size exclusion column using HBS. Purified proteins were made 20% in glycerol and stored at −80 °C.

## Purification of EaCDCLs for transmission electron microscopy experiments and crystallization

EaCDCL$^L$ and EaCDCL$^S$ were purified as above[7] with the following modifications. EaCDCL$^L$ cell pellets were resuspended in lysis buffer (50 mM HEPES pH 7.0, 500 mM NaCl, 20 mM imidazole, 1 mM TCEP, 5% (v/v) glycerol, 0.01% (v/v) Triton X-100) supplemented with DNase I (Sigma-Aldrich), lysozyme (Astral Scientific) and protease inhibitors (Sigma-Aldrich). EaCDCL$^S$ cell pellets were resuspended in lysis buffer with 2 mM CaCl$_2$ and 5% (v/v) PEG 400. Following affinity chromatography, fractions corresponding to His-EaCDCL$^L$ or His-EaCDCL$^S$ were pooled, and buffer-exchanged into 20 mM HEPES pH 7.0, 150 mM NaCl, 5 mMCaCl$_2$, 5% (v/v) glycerol using a HiPrep 26/10 desalting column (Cytiva). The His-tag was removed by cleavage with thrombin (1 IU per mg of EaCDCL) overnight at 4 °C and further purified by passage over Ni$^{2+}$-IMAC. Pooled EaCDCL$^L$ was concentrated (30 K MWCO centrifugal filter, Amicon) and loaded onto a 16/600 Superdex 200 column (Cytiva) equilibrated in 20 mM HEPES pH 7.0, 50 mM NaCl, 1 mM TCEP, 5% (v/v) glycerol. Pooled EaCDCL$^S$ was concentrated using a Vivaspin Turbo-15 10 K MWCO PES ultrafiltration unit (Sartorius) and passed over the Superdex 75 column (Cytiva) equilibrated in 20 mM HEPES pH 7.5, 150 mM NaCl, 2 mM CaCl$_2$, 5% (v/v) PEG 400. Fractions were stored at −80 °C.

## Crystallization of EaCDCL$^S$

Crystallization was conducted with EaCDCL$^S$ protein at 5.26 mg/mL in 20 mM HEPES pH 7.5, 150 mM NaCl, 2 mM CaCl$_2$, 5% (v/v) PEG 400. Crystallization was carried out using a sitting drop method in 96-well 3 lens low profile plates (SWISSCI) with a Gryphon liquid dispensing robot (Art Robbins). Crystallization drops contained 0.2 μL EaCDCL$^S$ and 0.2 μL of reservoir solution (0.1 M Bis-Tris pH 6.4, 0.1 M sodium chloride, 1.75 M ammonium sulfate) with 45 μL of crystallization condition in the reservoir at 22 °C. Crystals were briefly soaked in cryoprotectant solution (reservoir solution with 20% (v/v) glycerol) prior to cryocooling in liquid nitrogen.

## Crystal structure determination of EaCDCL$^S$

X-ray diffraction data were collected at the MX2 Beamline at the Australian Synchrotron at 100 K and a wavelength of 0.9537 Å using the ACRF Eiger 16 M Detector. EaCDCL$^S$ crystals diffracted to a maximum resolution of 1.85 Å resolution. Diffraction data were processed with DIALS using xia2[30,31] and AIMLESS[32] from the CCP4 software suite, using the ccp4i2 interface[33]. Crystals belonged to the space group $P3_2 21$ with unit cell dimensions of $a = b = 171.6$ Å, and $c = 61.3$ Å. The structure was determined by molecular replacement using Phaser[34]. The initial search model included D1, D2, and D3 of the EaCDCL$^L$ crystal structure (PDB ID: 6XD4, residues 34−367) and was prepared using *Sculptor* in *Phenix*[35] with completeness-based similarity deletion of the main-chain and similarity-based pruning of side-chain. The initial model was subject to automated building and refinement using AutoBuild in *Phenix*[35] before further manual modification in COOT[36] and refinement using *Phenix.Refine*[37] The final model geometry was analyzed using MolProbity[38]. The final model consists of two molecules of EaCDCL$^S$, 398 water molecules, and multiple ligands, including 2 sulfate ions, 2 calcium ions, 1 sodium ion, 3 glycerols, two 1.2-ethanediol and one tetraethylene glycol, all of which can be found as components of the buffer solution, crystallization condition or

cryoprotectant. The model has been deposited to the Protein Data Bank under the accession code 8G32. Supplementary Table 1 contains the data collection and refinement statistics for the crystal structure.

## Electron microscopy with gold-labeled EaCDCL[L]

Purified EaCDCL[S] was cleaved with trypsin at a ratio of 1:100 (w/w) for 5 min at room temperature then stopped with 1 mM PMSF. The sample was purified using a Superdex 75 16/600 column equilibrated in 20 mM HEPES pH 7.5, 150 mM NaCl, 2 mM CaCl₂, 5% (v/v) PEG 400. Fractions corresponding to activated-EaCDCL[S] were pooled and analyzed by intact protein LC-MS to confirm cleavage at the expected site (K[87]) in the activation loop, before storage at −80 °C. Purified EaCDCL[L] was cleaved with proteinase K (New England Biolabs) at a ratio of 1:300 (w/w) for 5 min at room temperature. The reaction was stopped with 1 mM PMSF, and the sample passed over a Superdex 200 16/600 column equilibrated in 20 mM HEPES pH 7.0, 50 mM NaCl, 1 mM TCEP, 5% (v/v) glycerol. The activated proteins were stored at −80 °C.

For gold labeling, a mutant of EaCDCL[L] (C347A/N239C) was used where the native cysteine residue was removed (C347). This derivative was expressed, purified and activated as above for EaCDCL[L]. The protein was labeled using a 5 nM Maleimide-Activated Gold Nanoparticle Conjugation Kit (Cytodiagnostics) per manufacturer's instructions. Activated-EaCDCL[L, C347A/N239C] was labeled for 1 hr at room temperature and unlabeled protein removed using a 100 K MWCO centrifugal filter. The gold-conjugated protein was stored at −80 °C until use.

For TEM experiments, a custom Teflon block was used. Gold-labeled activated-EaCDCL[L] (0.15 or 0.4 μM) and activated-EaCDCL[S] (0.3 or 0.4 μM) were added to a pre-chilled Teflon well on ice. The stoichiometries trialed were based on previous observations that decreasing the ratio of EaCDCL[L] to EaCDCL[S] resulted in altered kinetics of pore formation, A POPC lipid solution (1 μL at 0.5 mg/mL in chloroform) was applied to the well solution and allowed to evaporate to form lipid monolayers. A formvar/carbon-coated square-mesh (mesh size 300) grid was applied to the monolayer, and the entire Teflon block incubated at 37 °C for 5–25 minutes. The grid was removed and stained with 2% uranyl acetate before imaging. Imaging was conducted using a FEI Talos L120C transmission electron microscope (Thermo Fisher Scientific) at 120 keV or a Tecnai G2 F30 (FEI) at 200 keV. Microscopes were fitted with a CETA 4 × 4k CMOS camera. Images were collected at a nominal magnification of 59,000x at ~1.0–2.0 μm underfocus.

## Construction of integrative expression vectors and immunity gene expression in heterologous strains

All primers used in this study are listed in Supplementary Table 2. All plasmids created in this study were verified by whole plasmid sequencing. Phusion polymerase (NEB) was used to amplify all PCR products for cloning, and NEBuilder (NEB) was used to join all DNA pieces. pNBU2-*bla-ermG-* based[39] expression vectors were created so that the *B. fragilis* YCH46 immunity gene (BF1274) could be expressed as a single copy and integrated into relevant genomes. The promoter region of pFD340[40] was PCR amplified with the primers listed in Supplementary Table 2 and cloned into the BamHI site of pNBU2-*bla-ermG* and transformed into *E. coli* S17 λ pir creating pKF35. To swap *ermG* of pKF35 with *cfxA* to confer cefoxitin resistance instead of erythromycin resistance, *cfxA* was amplified with its promoter from *P. vulgatus* CL11T00C01[41] and pKF35 was amplified so that *ermG* was removed. These two DNA pieces were joined and transformed into S17 λ pir. The immunity gene BF1274 was amplified from *B. fragilis* YCH46 and cloned into the BamHI site of pKF35 and pKF54. The immunity gene construct in pKF35 or pKF35 vector were conjugally transferred from *E. coli* S17 λ *pir* into *P. vulgatus* CL09T03C04 and the immunity gene construct in pKF54 or pKF54 vector were conjugally transferred into *P. dorei* 9_1_42FAA. Transconjugants were selected on BHIS plates with

gentamycin (200 μg/ml) and erythromycin (10 μg/ml) or cefoxitin (10 μg/ml).

## Deletion of BF1274 (immunity gene) from *B. fragilis* YCH46

The DNA regions flanking BF1274 were PCR amplified and cloned into BamHI digested pMLS36[42] and transformed into *E. coli* S17 λ pir. The resulting plasmid was conjugally transferred to *B. fragilis* YCH46 and transconjugants were selected on BHIS plates containing gentamycin and cefoxitin. The cointegrant was passaged in non-selective medium and plated on BHIS plates with anhydrotetracycline (75 ng/ml). Double cross-over recombinants were screened for the mutant genotype by PCR.

## CF-loaded liposome preparation

CF-loaded POPC or POPC-cholesterol liposomes were prepared as previously decribed[43]. For the POPC liposomes 25 mg of POPC was dried under argon and then further dried overnight under vacuum. The lipid was resuspended by stirring in 4 ml of HBS and then freeze-thawed in liquid nitrogen four times. The lipid mixture was passed 21 times through a 100μm filter in a mini extruder (Avanti Polar Lipids). The liposomes were then passed over a column (1.5 × 20 cm) packed with Sephadex G-50 equilibrated in HBS to remove unentrapped CF. The liposomes were then stored on ice until used. The same procedure was used for the POPC-cholesterol-liposomes, except a mixture of 25 mg POPC was mixed with 15.5 mg of cholesterol.

## Liposome marker release assays

The following reactions were carried out as individual experiments performed in triplicate as follows: 16.4 μM CF-POPC liposomes, either 700 nM BfCDCL[L] or EaCDCL[L], and the corresponding BfCDCL[S] or EaCDCL[S] at 0.25, 0.5, 1, 5, 10, 15, 20, 25, and 30 molar ratios relative to either BfCDCL[L] or EaCDCL[L]. The final volume of all reactions was brought to 100 μL using HBS. Trypsin from bovine pancreas (Sigma-Aldrich) was used to cleave the CDCLs at a 15:1 CDCL:trypsin ratio (w/w) for 30 minutes at room temperature and then diluted to a final volume of 3 mL in HBS. All additional controls were performed using the same procedure. Fluorescence measurements were made using a Fluorolog-3 Spectrofluorometer with the FluorEssence software (excitation at 485 nm/emission at 520 nm). Measurements were taken until the standard error of the mean between all the measurements was below two percent. All data was blank-corrected to remove intrinsic fluorescence of the CF-loaded liposomes and unlabeled protein and was plotted using Prism 8.0.1 (Graphpad).

## Kinetic marker release assays from liposomes

For the kinetic release assays a master mix for each assay was prepared containing the following: 290 nM CDCL[L], 2.9 μM CDCL[S], 18.8 μM CF-POPC or POPC-cholesterol liposomes and brought to a total volume of 1 mL with HBS and incubated in a 37 °C water bath for 5 minutes. 175 μL aliquots were distributed into 4 wells of a Greiner 96-well plate and placed into a BMG FLUOstar Omega plate reader equipped with dual injector pumps that were set to bottom optics emission acquisition (excitation at 485 nm, emission at 520 nm), 30 flashes per well, and the chamber was maintained at 37 °C. The CF emission was monitored for 2 minutes prior to automated injection of HBS or protease (dependent on the experiment) to establish a baseline. Then either 25 μL HBS or 5.25 μM trypsin was injected into each sample. The plate was shaken for 5 seconds after injection. The final concentration after injection is 250 nM CDCL[L], 2.5 μM CDCL[S], 16.4 μM CF-POPC liposomes, and 651 nM trypsin. For all kinetic experiments, the emission data was normalized to 100 arbitrary units to the emission resulting from the positive control wherein the assay only contained activated-EaCDCL or BfCDCL and liposomes.

## FRET experiments

The CDCL derivatives (EaCDCL[L-C347A/N239C], EaCDCL[S-N248C], BfCDCL[L-C31S/C32S/N227C], BfCDCL[SK361C]) were labeled with either a donor (D, AZDye 488 maleimide, Fluoroprobes, Inc.) or acceptor (A, Tetramethylrhodamine-5-maleimide, Fluoroprobes, Inc.) fluorescent dye and then combined with the unlabeled (U) derivative as background controls. Donor emission alone in the presence or absence of liposomes was corrected for any light scattering, autofluorescence, or direct excitation of the fluorescently labeled proteins by subtracting the U emission from the DU emission. For samples where both D and A-labeled protein was present, the UA emission was subtracted from the DA emission to correct for any contribution of the A to the emission. Donor quenching was then determined by comparing the emission of the corrected DU to the corrected DA emission[18]. For samples containing CDCL[L] and CDCL[S] as either D or A-labeled protein, a 1:10 molar ratio of CDCL[L] (250 nM) to CDCL[S] (2.5 μM) was used. For samples containing CDCL[L] as D and A plus unlabeled CDCL[S], a four-molar excess of A-labeled protein was used compared to D-labeled protein. In those samples, total protein was at a molar ratio of 1:10 of CDCL[L] to CDCL[S]. For samples containing CDCL[S] as D and A plus unlabeled CDCL[L], a four-molar excess of A-labeled protein was used compared to D labeled protein. In those samples, total protein was at a molar ratio of 1:10 of CDCL[L] to CDCL[S]. For samples containing either CDCL[L] only (250 nM) or CDCL[S] only (2.5 μM), again, a four-molar excess of A-labeled protein was used compared to D labeled protein. For activation trypsin was included in the sample at a 1:15 w/w ratio to the total CDCL protein. If POPC liposomes were included in the sample, the equivalent of 16.4 μM of lipid was added. All samples were brought to a total volume of 50 μL in HBS and then incubated at room temperature (EaCDCLs) or at 37 °C (BfCDCLs) for 30 minutes in the dark. After incubation, all samples were brought to a final volume of 200 μL in HBS and placed into a 96-well plate (Corning 3615). The emission of the donor fluorophore was recorded using a FLUOstar Omega Series Plate Reader (BMG Labtech) programed with an excitation wavelength of 485 nm and an emission wavelength of 520 nm, using bottom optics and 30 flashes per well. After the mathematical corrections were made as mentioned above, DU emission was normalized to 100 and DA emission was calculated as a percentage of DU.

## Inhibition assay of CDCL pore-forming activity by BcdI

For experiments involving the BcdI inhibition of BfCDCL pore-forming activity, fragipain was substituted for trypsin. For the BcdI assays, a master mix of each assay was prepared containing the following: 290 nM BfCDCL[L], 2.9 μM BfCDCL[S], 18.8 μM CF-POPC liposomes, and brought to a total volume of 875 μL using HBS. Additional master mixes were created which contained BcdI at a 0.1:1, 0.3:1, 0.5:1, and 1:1 molar amount relative to BfCDCL[S]. A set of control mixes were generated using bovine serum albumin for the BcdI. The master mixes were incubated in a 37 °C water bath for 5 minutes and 175 mL aliquots were distributed into 4 wells of a Greiner 96-well plate, one of which served to blank out the intrinsic fluorescence of the liposomes. Baseline fluorescence emission was recorded for 2 min then 25 μL HBS was injected into the blank and 25 μL of 4.28 μM fragipain was injected into each sample to initiate the activation of the BfCDCL and shaken for 5 seconds. The final concentration after injection of the fragipain or HBS was 250 nM BfCDCL[L], 2.5 μM BfCDCL[S], 16.4 μM CF-POPC liposomes, and 554 nM fragipain. For each of the experimental groups the concentrations of BcdI were 0.25 μM, 0.75 μM, 1.25 μM, and 2.5 μM. BcdI impact on pore formation by the BfCDCL was also determined four minutes after injection of fragipain to initiate the activation of toxin or HBS 25 μL of either BSA or BcdI at a 0.3:1 ratio relative to BfCDCL[S] was injected. The final concentration of the constituents after the final injection is 250 nM BfCDCL[L], 2.5 μM BfCDCL[S], 16.4 μM CF-POPC liposomes, and 554 nM fragipain. Fluorescence measurements were performed using a BMG FLUOstar Omega plate reader (excitation at 485 nm/emission at 520 nm). Assays were performed in triplicate using bottom optics, 30 flashes per well, and were carried out at 37 °C. Data was normalized and plotted using Prism 8.0.1 (GraphPad) for both kinetic and endpoint analysis.

## Growth inhibition assays

Bacteroidales species were grown in basal medium[44] under anaerobic conditions at 37 °C to an $OD_{600}$ of ~0.4 and 10 μL of the culture was added to 125 μL fresh broth in a 96-well plate and incubated for 30 minutes before 15 μL activated (1:1 w/w fragipain:BfCDCL for 30' at room temperature prior to addition) or unactivated BfCDCL toxin was added to a final concentration of 250 nM BfCDCL[L] and 2.5 μM BfCDCL[S]. All experiments were individual growth curves performed in triplicate. Either HBS or fragipain alone were added to control wells. Readings ($OD_{600}$) were recorded every hour using an Eon high-performance microplate spectrophotometer (BioTek Instruments, Winooski, VT) or using a Cerrillo microtiter plate reader (Cerrillo) and plotted as the standard error of the mean using Prism 8.0.1 (GraphPad).

## PI analysis of BfCDCL-treated cells

For time-course imaging of toxin activity, *P. dorei* 9_1_42FAA cells were grown anaerobically to mid-exponential phase in basal medium. The BfCDCL[S] and BfCDCL[L] were mixed at a 10:1 ratio and added to the cells at a final concentration of 2.5 μM BfCDCL[S]:250 nM BfCDCL[L]. At each time point, cells were dotted onto M9 minimal medium 1.5% low-melt agarose pads with 15 μM PI and imaged immediately. Differential interference contrast (DIC) and fluorescent images were acquired on an Olympus IX81 inverted widefield microscope with a 100 × 1.45 oil immersion objective, using a Hamamatsu Orca Flash 4.0 digital camera and SlideBook imaging software. PI signal was detected in the red channel using emission/excitation filters 530-550 and 590LP with a 100 ms exposure time. Image overlays were generated using Fiji 2.13.1[45], with the PI signal normalized to minimum and maximum displayed values of 7498 and 54,256, respectively. Image segmentation to quantify fluorescent PI signal for each cell was calculated using MicrobeJ 5.13I[46]. A minimum of three frames were analyzed, with a minimum 555 cells per time point. Mean intensity across the cell area in the red channel was computed, and cells were counted as PI-positive if they had a value above the background of 7498.

## Reporting summary

Further information on research design is available in the Nature Portfolio Reporting Summary linked to this article.

# Data availability

Accession codes for all proteins analyzed in for this study can be found in Supplementary Dataset 1. Accession codes for the metagenomic datasets analyzed in the present study can be found in Supplementary Dataset 2. All protein structure PDB codes and AlphaFold2 database derived structures have links in the paper. The crystal structure data for the EaCDCL[S] generated in this study has been deposited in the Protein Data Bank under accession code 8G32. Source data are provided with this paper.

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

## Acknowledgements

We thank P. Parrish for technical help in purifying proteins and Dr Sara Lawrence, who grew the first crystals of EaCDCL$^s$. This work was supported by a US National Institutes of Health grant (NIAID) 5R37AI037657-27 to R.K.T. L.E.C. is supported by the Duchossois Family Institute and US National Institutes of Health grant (NIAID) R01AI093771. B.A.J. was the recipient of an Australian Government Research Training Program (RTP) Scholarship. This work was also supported by Australian Research Council Discovery Projects grants (DP200102871, DP230101148) to M.W.P. and C.J.M. Infrastructure support from the National Health and Medical Research Council of Australia (NHMRC) Independent Research Institutes Infrastructure Support Scheme and the Victorian State Government Operational Infrastructure Support Program to St. Vincent's Institute are gratefully acknowledged. M.W.P. is an NHMRC Leadership Fellow (APP1194263). L.G.-B. is supported by the US National Institutes of Health grant (NIAID) K99AI167064. This research was partly undertaken at the Australian Synchrotron, part of the Australian Nuclear Science and Technology Organization, on the MX beamlines and made use of the ACRF Detector on the MX2 beamline. Electron microscopy was performed at the Ian Holmes Imaging Center (IHIC), located at the Bio21 Molecular Science and Biotechnology Institute, and we thank the technical support of facility staff.

## Author contributions

H.L.A., T.C.S., C.E.C., and B.A.J. contributed equally to the work done in this manuscript and the design of experiments. H.L.A. designed and performed endpoint analysis and identified and analyzed the function of the immunity protein. L.G.-B. performed the microscopy and PI experiments, T.C.S. designed and performed the in vitro and in vivo studies associated with proteolytic activation of the CDCLs. C.E.C. designed and performed the FRET studies. B.A.J. solved the structure of the EaCDCL$^S$ and, together with M.P.C., performed electron microscopy studies. C.J.M. provided advice on the structural studies. K.F. made cloning vectors for immunity gene expression. L.E.C. made genetic constructs performed some of the growth inhibition studies, and contributed to writing the manuscript. J.C.E. performed early experiments that indicated a MAC-like mechanism of the CDCLs. A.J.F. generated and purified some derivatives of the EaCDCLs used in these studies. M.J.C. performed the genomic and metagenomic analyses of the CDCL family proteins. M.W.P., L.E.C., and R.K.T. supervised the work done in this study. R.K.T. performed early studies on the activation of the CDCLs and wrote the manuscript with input from all authors.

## Competing interests

R.K.T. has a provisional patent application (serial no. 63/506,273) on engineering the CDCLs to redirect their pore formation towards other prokaryotic cells and eukaryotic cells. The remaining authors declare no competing interests.
