## [Peer Review File · Nature Communications]

Distant relatives of a eukaryotic cell-specific toxin family evolved a complement-like mechanism to kill bacteriaReviewer #1 (Remarks to the Author):

The manuscript by Hunter and colleagues describes functional properties of a family of pore-forming toxins from *Bacteroides*, CDCL. These toxins are similar to well-known cholesterol-dependent cytolysins, but act by distinctively different mechanism and could represent a new paradigm for pore-forming proteins in general. The specifics include proteolytic activation at the surface of the bacterial cell, requirement of two different components for efficient pore formation and possibility to inhibit the activity by a surface protein, which is employed by producing cell for its protection. This is an interesting and novel topic and will be of interest for a wider scientific community.

The following comments may further improve the manuscript.

Page 4: It is stated in the introduction that the authors previously reported finding CDCL in *Bacteroidota* species. Please cite this reference or work, it is not clear which of the cited references (11-14) refer to large and small CDCL components.

Page 7: It is claimed that the authors previously solved crystal structure of CDCLL from *E. anophelis*. Please cite these data and reference.

Page 7: the text needs to reflect that AlphaFold models of *B. fragilis* are available to the authors at the moment and not experimentally determined structures (i.e. line 8 from below).

Page 9: Please comment, from the structural point of view, what is being removed by the proteolytical cleavage of CDCL components and how the protease activation affects the structure and consequently enabled the permeabilizing activity. This requires a bit more attention and explanation as it is one of the crucial new features of this system.

Page 11: can receptors for CDCLL be discussed? What could be the range of molecules on the surface of bacteria to which CDCLL could bind? Data in Figure 3a indicate that this could be lipid, as the activity assays were done on liposomes in absence of any protein.

Page 12, Figure 4a: The lines for activated BfCDCL (the blue line) are missing from some graphs on Figure 4. Please add or show differently if it is not visible properly.

Page 13: the panels that describe PI staining (Fig 4b and 4c) are not properly referred to in the text (first line). In line 8, a panel that shows activity of BfCDCL cleavage site mutants is not referred to correctly (it is Figure 5a).

Page 13: The data should be presented (for example gels) that cleavage site mutations R70A and R62A for BfCDCLL and BfCDCLS, respectively, indeed prevent cleavage of the proteins by proteases.

Page 14: The finding: "Furthermore, when the gene encoding this lipoprotein is deleted in the BfCDCL producing strain *B. fragilis* YCH46, it becomes more sensitive to its own CDCL pair (activated and unactivated)." should have a citation to the panel (Fig. 5d). Why is there no difference in activity of activated and unactivated CDCL?

Figure 1: it is stated in the legend that regions where the two MGE diverge are shown with vertical lines. I miss them on the figure, please check.

Figure 2: Please add PDB codes to structures, where applicable, in the legend to Figure 2.

Figure 3: please state, in the legend to figure, the number of experiments and what error bars represent on panel a. Also, please state the number of experiments and show variability between the repetitions in panel b (is a single measurement represented by the data?).

Figure 4a and Figure 5: All growth curves should be quantified. Only one curve is shown for each condition and no indication about variations between the experiments is indicated (it is stated that

all experiments were performed twice in triplicate wells in the experimental section. Information about the number of experiments should be provided in the legends to figures). For example, data in Figure 5D show reduced but significant activity at some of the experimental combinations, but not the same to controls.

Figure 4c: I do not understand box plots. What parameters are represented by the horizontal line, the box and the whiskers? Could data points also be shown? Why box disappears in some time points? If I understand correctly the experiment, then this representation of the data may not be the most appropriate (Are three-five points (i.e. separate fields as stated in the legend) used per time point? Why not using column plot?).

Figure 6: please check the text in the legend to figure 6 (the sentence in lines 4-5).

Supplemental Figure 6: Some experimental details are missing in the legend to figure and should be reported, for example, what was the concentration of the protein in the assay, buffers used etc.

Reviewer #2 (Remarks to the Author):

Overall assessment:

The study by Hunter et al., presents an exciting new finding in the field of pore-forming proteins. They functionally characterize a 2-component toxin system secreted by *Bacteroides* to kill other bacterial species in the battle for maintaining gut microbiome. Building on their previous crystallographic studies of a long-version of this toxin CDCLI, they now solve the structure of the short version CDCLs. Using a series of biochemical and biophysical assays they show that CDCLI initiates membrane binding and is proteolytically cleaved to enable incorporation of CDCLs and propagation of the pore. I have a few comments that would improve the readability of the article and discussion points to consider which will increase the impact and breadth of the study.

Introduction:

There is a sentence here that refers to unpublished data. This statement should either be removed or the underlying data supporting it made available in the supplemental material.

Results:

Fig 1. The domains of C9 and C8a should be similarly colored given the focus of the figure is emphasising structural conservation. One question. Why choose to color the LDRA domain of C9 green? Are you trying to make the reader think about similarities with the b-tongue of the bacterial pore forming proteins which is also colored green? More likely it is the EGF domain that is having a similar function in the latch moving to release the TM hairpins of C9.

Fig. 3. In the negative stain image of your pores formed on lipid monolayers, why do you see size views? A similar comment for Supplementary Fig. 5. Also, I found it confusing to quickly understand what was being shown by the numbered boxes in the micrograph. At first glance it isn't easy to tell they are a magnification of the box, as they appear embedded in the micrograph. Better to box in the micrograph and below line up the enlargements.

Figure 4. Please add the scale bar value in the legend.

Discussion:

The discussion does a great job drawing parallels of this system with the MAC initiation and inhibition mechanisms. I have two suggestions that might broaden the scope of this study.

1)What was the alphafold2 prediction of the protector protein BcdI? Were there any similarities to the structure of CD59?

2)MPEG-2 is also proteolytically cleaved to activate pore formation. This could be brought into the discussion.

Reviewer #3 (Remarks to the Author):

This manuscript from Abrahamsen et al. describes the biochemical characterization of 2 different pore-forming bacterial toxins, termed CDCL (CDC-like) toxins. These are 2-component toxins comprised of large (CDCLL) and small (CDCLS) subunits. The authors found, unexpectedly, that these are more reminiscent of eukaryotic membrane attack complexes (MAC) than bacterial CDCs. The authors use well-planned and well-executed experiments to conclusively and comprehensively show that CDCLs from *E. anopheles* and *B. fragilis* bind an unknown receptor on bacterial rather than eukaryotic cells, and require proteolytic activation by the C11-type protease DpnB for efficient pore formation. They also describe the discovery of an immunity protein, called BcdI that protects *B. fragilis* from its own CDCLs.

Overall this manuscript is very well-written and represents an important paradigm-changing advance in the field. I recommend publication of this manuscript with very minor revisions, described below.

1) In the introduction, at the end of the first paragraph, the authors highlight unpublished data. Can these data be included in this manuscript?

2) As shown in Fig. 3a, both EaCDCL and BfCDCL show activity on POPC liposomes upon proteolytic activation by trypsin. However, when the putative receptor binding domain from EaCDCLL was replaced with D4 from PFO, this chimera was only active on POPC-cholesterol (not POPC) liposomes, which demonstrates the expected targeting of cholesterol by PFO D4. Does this suggest that native EaCDCL and BfCDCL bind phosphocholine as a receptor, since the activities in Fig 3a and Suppl Fig 6 look similar?

3) In figure 2, it might help if the authors could present the AlphaFold structure coloured by the Predicted local distance difference test (pLDDT) from very low to very high to see which domains are best conserved.

4) In reference to Figure 5d, the authors state in the results section: "Next, we cloned and expressed it in sensitive cells and found that they become largely resistant to the activated and unactivated BfCDCL pairs"....Can the authors expand in the results/discussion on this - specifically the magnitude of the effects they see in the left two panels (blue curve) - the partial rescue. Is it a matter of expression of BfCDCL? Would be interesting to see if they can ablate it as they do above. Not critical to do an extra experiment here, but I think it deserves a comment.

REVIEWER COMMENTS

Reviewer #1 (Remarks to the Author):

The manuscript by Hunter and colleagues describes functional properties of a family of pore-forming toxins from Bacteroides, CDCL. These toxins are similar to well-known cholesterol-dependent cytolysins, but act by distinctively different mechanism and could represent a new paradigm for pore-forming proteins in general. The specifics include proteolytic activation at the surface of the bacterial cell, requirement of two different components for efficient pore formation and possibility to inhibit the activity by a surface protein, which is employed by producing cell for its protection. This is an interesting and novel topic and will be of interest for a wider scientific community.

The following comments may further improve the manuscript.

Page 4: It is stated in the introduction that the authors previously reported finding CDCL in Bacteroidota species. Please cite this reference or work, it is not clear which of the cited references (11-14) refer to large and small CDCL components.

We have placed the appropriate reference #15 at the end of the sentence “We previously reported that *Elizabethkingia anophelis*, a bacterial species of the phylum Bacteroidota and a commensal of the malarial mosquito midgut¹⁻⁴ produces a 2-component CDC-like (CDCL) set of proteins, encoded by adjacent genes: one large (CDCL^L) and one smaller component (CDCL^S)¹⁵”.

*Page 7: It is claimed that the authors previously solved crystal structure of CDCLL from *E. anophelis*. Please cite these data and reference.*

We have included the appropriate reference (#15) at the end of the sentence on page 7 that reads “We previously solved the CDCL^L crystal structure from *E. anophelis* (EaCDCL^L)¹⁵...”

*Page 7: the text needs to reflect that Alphafold models of *B. fragilis* are available to the authors at the moment and not experimentally determined structures (i.e. line 8 from below).*

We have changed the sentence to reflect these structures were available to us and updated the reference to that for the Alphafold database by stating “The *B. fragilis* BfCDCL^L and BfCDCL^S structures were available in the Alphafold database¹⁸ (designations BF1276 and BF1275, respectively) and resemble their *E. anophelis* analogues (Fig. 2).”

Page 9: Please comment, from the structural point of view, what is being removed by the proteolytical cleavage of CDCL components and how the protease activation affects the structure and consequently enabled the permeabilizing activity. This requires a bit more attention and explanation as it is one of the crucial new features of this system.

This was a great suggestion, as we had overlooked its explanation for readers. We have included a short paragraph on this topic in the second to last paragraph of the discussion (shown below). We have also modified Fig. 6 to add a panel (Fig. 6b) to help with the visualization of how proteolytic cleavage facilitates the interaction of the CDCL components with additional explanation below.

“As shown in Fig. 6b, cleavage of the CDCL^L propeptide likely allows β 5 (Fig. 6b, blue strand) to be displaced from its interaction with β 4 (Fig. 6b, red strand), which is the interface that interacts with β 1 (Fig. 6b, cyan strand) of the first incoming monomer of CDCL^S. The propeptide of the CDCL^S blocks β 1, which must be exposed to interact with β 4 of CDCL^L and of other CDCL^S as the oligomerization of the CDCL^S that form the β -barrel pore.”

Additional to Fig. 6 for panel (b)

“(b) Shown are the crystal structures of EaCDCL^L, EaCDCL^S and PFO. Upon proteolytic activation of CDCL^L the propeptide (blue and dashed line) may be lost but like the β -tongue of PFO^{21,70} (blue), this action may also be conformationally coupled to weaken the interaction of β 5 (blue) with β 4 (red), as we have found in PFO^{21,70}. Next, the CDCL^S propeptide must be cleaved and lost, or displaced, so that its β 1 (cyan) can form backbone hydrogen bonds with β 4 of CDCL^L and as more CDCL^S enters the assemble they must also form intermolecular interactions between β 4 and β 1. The same interactions take place between PFO monomers but displacement of β 5 from β 4 is trigger by membrane binding, which is conformationally coupled via the β -tongue (blue)^{21,70}.”

Page 11: can receptors for CDCLL be discussed? What could be the range of molecules on the surface of bacteria to which CDCLL could bind? Data in Figure 3a indicate that this could be lipid, as the activity assays were done on liposomes in absence of any protein.

As much as we would like to speculate on the receptors we feel that any discussion of the receptors would not enhance the manuscript, would add length to the manuscript and be purely speculative since we have no indication of they might be (we are working hard on this problem). Reviewer 3 also commented on this topic. We don't know why the CDCLs reported herein bind and form pores on the POPC liposomes since they do not do so on erythrocytes (see hemolytic data in supplementary Fig. 6b, which are rich in POPC. Hence, there is a deeper enigma about this interaction with the liposomes that we do not understand, and to complicate this further, POPC is not typically present in prokaryotic membranes. The fact that they are active on the liposomes is a convenient assay system but why the CDCL^L does so remains unclear and does not really provide any insight into possible receptors.

Page 12, Figure 4a: The lines for activated BfCDCL (the blue line) are missing from some graphs on Figure 4. Please add or show differently if it is not visible properly.

This was because once we determined that the *Phocaeicola* species were susceptible to unactivated BfCDCLs we only treated the three strains in the third row with unactivated CDCLs. We had indicated this in the figure legend but have added to that sentence to clarify this as show below. We also added a separate legend in the first graph in row 3 to further clarify that the cells were only treated with HBS or unactivated toxin.

“In the third row the strains were only treated with unactivated BfCDCLs.”

Page 13: the panels that describe PI staining (Fig 4b and 4c) are not properly referred to in the text (first line). In line 8, a panel that shows activity of BfCDCL cleavage site mutants is not referred to correctly (it is Figure 5a).

Good catch! We have corrected these errors (and a few others).

Page 13: The data should be presented (for example gels) that cleavage site mutations R70A and R62A for BfCDCLL and BfCDCLS, respectively, indeed prevent cleavage of the proteins by proteases.

We have included the gels for these mutants in supplementary Fig. 8. Note that the DpnB N-terminal fragment is highly sensitive to degradation during purification whereas the C-terminal fragment is not, therefore it is not as active as we would like but still cleaves at the same site as identified *in vivo*.

Page 14: The finding: “Furthermore, when the gene encoding this lipoprotein is deleted in the BfCDCL producing strain B. fragilis YCH46, it becomes more sensitive to its own CDCL pair (activated and unactivated).” should have a citation to the panel (Fig. 5d). Why is there no difference in activity of activated and unactivated CDCL?

We have added the figure reference. The reason that activated and unactivated exhibit the same activity against the *B. fragilis* immunity protein knockout is that the unactivated BfCDCLs are apparently activated by a protease by a surface protease of *B. fragilis*, which is likely the C11 protease of *B. fragilis*, fragipain. Therefore, both are equally active. We added the following sentence to clarify this.

“Both activated and unactivated exhibit the same effect, as the unactivated is apparently activated by a surface protease on the surface if *B. fragilis*.”

Figure 1: it is stated in the legend that regions were the two MGE diverge are shown with vertical lines. I miss them on the figure, please check.

We have boxed the regions and changed this in the legend.

Figure 2: Please add PDB codes to structures, where applicable, in the legend to Figure 2.

We have added PDB codes for the crystal structures of the EaCDCLs, the complement proteins and the AlphaFold database codes for the BfCDCLs to Fig. 2.

Figure 3: please state, in the legend to figure, the number of experiments and what error bars represent on panel a. Also, please state the number of experiments and show variability between the repetitions in panel b (is a single measurement represented by the data?).

We have added a sentence for Fig. 3a that these experiments were the average from 3 individual experiments and the error bars represent the standard error for each experiment.

I added the explanation of the data for panel (b) as shown below. Also, I should note that the FRET data was confirmed by the EM data in panel (c).

“Each pair of bars in the spectroscopic data is the average of experiments that were performed triplicate and are representative of more than 2 individual experiments. The emission data of the D+U and D+A pairs were normalized to the average of the highest emission of the D+U samples and the highest emission was set to 100 arbitrary units, as done previously^{27,67,68}.”

Figure 4a and Figure 5: All growth curves should be quantified. Only one curve is shown for each condition and no indication about variations between the experiments is indicated (it is stated that all experiments were performed twice in triplicate wells in the experimental section. Information about the number of experiments should be provided in the legends to figures). For example, data in Figure 5D show reduced but significant activity at some of the experimental combinations, but not the same to controls.

We have added the error bars to all growth curves in figures 4 and 5. In nearly all cases the error is so small that the error bars are not much bigger than the line representing the average of all three experiments.

Figure 4c: I do not understand box plots. What parameters are represented by the horizontal line, the box and the whiskers? Could data points also be shown? Why box disappears in some time points? If I understand correctly the experiment, then this representation of the data may not be the most appropriate (Are three-five points (i.e. separate fields as stated in the legend) used per time point? Why not using column plot?).

We have provided an explanation of the standard box plot in Fig. 4c shown below.

“The data are presented as a standard box plot wherein the boxes represent the quartile distribution of the PI-stained cells and the horizontal line represents the data median. The whiskers represent the highest and lowest number of PI stain cells from all 5 views.”

Figure 6: please check the text in the legend to figure 6 (the sentence in lines 4-5).

We have fixed this problem and streamlined the explanation.

Supplemental Figure 6: Some experimental details are missing in the legend to figure and should be reported, for example, what was the concentration of the protein in the assay, buffers used etc.

We added a sentence that the conditions and protein concentrations were the same as described in the methods for the assay of the native CDCLs: "Conditions and protein concentrations were as described for the native CDCLs in the Methods section that describes the kinetic marker release assays from liposomes."

Reviewer #2 (Remarks to the Author):

Overall assessment:

The study by Hunter et al., presents an exciting new finding in the field of pore-forming proteins. They functionally characterize a 2-component toxin system secreted by Bacteroides to kill other bacterial species in the battle for maintaining gut microbiome. Building on their previous crystallographic studies of a long-version of this toxin CDCLI, they now solve the structure of the short version CDCLs. Using a series of biochemical and biophysical assays they show that CDCLI initiates membrane binding and is proteolytically cleaved to enable incorporation of CDCLs and propagation of the pore. I have a few comments that would improve the readability of the article and discussion points to consider which will increase the impact and breadth of the study.

Introduction:

There is a sentence here that refers to unpublished data. This statement should either be removed or the underlying data supporting it made available in the supplemental material.

We have added the unpublished hemolytic data into supplemental Fig. 6 along with the activity of the EaCDCL^L whose binding domain was swapped with the PFO cholesterol binding domain: unlike the native EaCDCLs it is active on the erythrocytes, which was expected since it is active on cholesterol-POPC liposomes (supplemental figure 6).

Results:

Fig 1. The domains of C9 and C8a should be similarly colored given the focus of the figure is emphasizing structural conservation. One question. Why choose to color the LDRA domain of C9 green? Are you trying to make the reader think about similarities with the b-tongue of the bacterial pore forming proteins which is also colored green? More likely it is the EGF domain that is having a similar function in the latch moving to

release the TM hairpins of C9.

That was unintentional, we have changed the color to brown. The only colors that are for comparison between the complement proteins with the PFO and the CDCLs are the red and yellow α -helical bundles.

Fig. 3. In the negative stain image of your pores formed on lipid monolayers, why do you see size views? A similar comment for Supplementary Fig. 5. Also, I found it confusing to quickly understand what was being shown by the numbered boxes in the micrograph. At first glance it isn't easy to tell they are a magnification of the box, as they appear embedded in the micrograph. Better to box in the micrograph and below line up the enlargements.

We have done as suggested. For Supplementary Fig. 5 we see side views of activated-EaCDCL^S oligomers (eg. Supp Fig 5 panels a-b), as these do not require a lipid monolayer for formation instead forming spontaneously in solution. Therefore, they do not need to be orientated with regards to the lipid monolayer.

Figure 4. Please add the scale bar value in the legend.

We have added it to the first panel (10 μ m).

Discussion:

The discussion does a great job drawing parallels of this system with the MAC initiation and inhibition mechanisms. I have two suggestions that might broaden the scope of this study.

1)What was the alphafold2 prediction of the protector protein Bcd1? Were there any similarities to the structure of CD59?

There were no structural similarities with CD59 or with any other protein. Since we do not have any data in the immunity protein, other than it inhibits the assembly of a functional pore, we decided to leave an AlphaFold structure of the immunity protein out for now, as it would not add to the paper and any structure/function discussion would purely speculative.

2)MPEG-2 is also proteolytically cleaved to activate pore formation. This could be brought into the discussion.

We assume that the reviewer means Perforin-2 (MPEG1 gene), as there are no references to an MPEG2 in the literature. The proteolytic activation of Peforin-2 remains unclear in terms of its mechanism, but the available data suggests that it can form prepore oligomers on the macrophage endosomal membrane and that it is cleaved off its membrane anchor domain. How this affects activity is unknown, so it makes it difficult to compare with the activation of the CDCLs at the surface of sensitive cells.

Reviewer #3 (Remarks to the Author):

*This manuscript from Abrahamsen et al. describes the biochemical characterization of 2 different pore-forming bacterial toxins, termed CDCL (CDC-like) toxins. These are 2-component toxins comprised of large (CDCLL) and small (CDCLS) subunits. The authors found, unexpectedly, that these are more reminiscent of eukaryotic membrane attack complexes (MAC) than bacterial CDCs. The authors use well-planned and well-executed experiments to conclusively and comprehensively show that CDCLs from *E. anopheles* and *B. fragilis* bind an unknown receptor on bacterial rather than eukaryotic cells, and require proteolytic activation by the C11-type protease DpnB for efficient pore formation. They also describe the discovery of an immunity protein, called BcdI that protects *B. fragilis* from its own CDCLs.*

Overall this manuscript is very well-written and represents an important paradigm-changing advance in the field. I recommend publication of this manuscript with very minor revisions, described below.

1) In the introduction, at the end of the first paragraph, the authors highlight unpublished data. Can these data be included in this manuscript?

We have added the unpublished hemolytic data into supplemental Fig. 6 along with the activity of the EaCDCL^L whose binding domain was swapped with the PFO cholesterol binding domain: unlike the native EaCDCLs it is active on the erythrocytes, which was expected since it is active on cholesterol-POPC liposomes (supplemental figure 6).

2) As shown in Fig. 3a, both EaCDCL and BfCDCL show activity on POPC liposomes upon proteolytic activation by trypsin. However, when the putative receptor binding domain from EaCDCLL was replaced with D4 from PFO, this chimera was only active on POPC-cholesterol (not POPC) liposomes, which demonstrates the expected targeting of cholesterol by PFO D4. Does this suggest that native EaCDCL and BfCDCL bind phosphocholine as a receptor, since the activities in Fig 3a and Suppl Fig 6 look similar?

This is something we have thought about very hard, but the fact that the native EaCDCLs were inactive on erythrocytes (supplementary Fig. 6b), which are rich in POPC, suggested that there is something more to this interaction with the POPC liposomes that we cannot currently deconvolute.

3) In figure 2, it might help if the authors could present the AlphaFold structure coloured by the Predicted local distance difference test (pLDDT) from very low to very high to see which domains are best conserved.

We have included them as panel (b) in Fig. 2.

4) In reference to Figure 5d, the authors state in the results section: "Next, we cloned

and expressed it in sensitive cells and found that they become largely resistant to the activated and unactivated BfCDCL pairs"....Can the authors expand in the results/discussion on this - specifically the magnitude of the effects they see in the left two panels (blue curve) - the partial rescue. Is it a matter of expression of BfCDCL? Would be interesting to see if they can ablate it as they do above. Not critical to do an extra experiment here, but I think it deserves a comment.

We have considered this and there are no ways to increase expression over our approach (*Bacteroides* just don't have the genetic tools available to do this like *E. coli* and others). We suspect that there may be other factors involved in the resistance of *B. fragilis* to its own toxin when the immunity protein is knocked out (e.g., *B. fragilis* receptor might not be bound as well, the membrane composition might be different, etc). We have also commented on this for reviewer 1 in the following way by adding the following to the second paragraph of the Discussion.

"However, the loss of the BcdI from *B. fragilis* only exhibits a partial loss of protection suggesting there may be other factors that help protect it from its own toxin (Fig. 5d, left panel). Consistent with this observation, substantial but not complete protection was seen when the BcdI is introduced into sensitive strains (Fig. 5d, right 2 panels). This is not unexpected, as there may be other reasons *B. fragilis* was not completely sensitive to its own toxin, which remain to be revealed once the system is better understood. This is also true for the complement MAC, as multiple factors are also required for host protection from the MAC³⁸."

Reviewer #3 (Remarks to the Author):

The authors have addressed all of my comments/concerns.